# Transposon invasion of primate genomes shaped human inflammatory enhancers and susceptibility to inflammatory diseases

Mengliang Ye[1], Maxime Rotival [2], Sebastian Amigorena [1] & Elina Zueva [1] ✉

Human inflammatory response reflects adaptive alteration of immune-cell regulatory elements during human evolution. Yet the impact of the deeper evolutionary history of these elements, within primate genomes reshaped by transposon expansions, remains unclear. Tracing sequence changes in human immune-cell enhancers back to macaque and analysing proinflammatory transcription factor binding, we show that primate-specific endogenous retroviruses and Alu transposons introduced functional NF-κB and IRF1 motifs, contributing most to the great-ape–specific pool. After the human-macaque split, these motifs tend to evolve toward higher predicted binding affinity. In modern humans, positive selection favoured alleles, often Alu-derived, that increase enhancer affinity for NF-κB, and Alu-containing enhancers are enriched in signatures of adaptation. Highly mutable, Alus disproportionately contribute to the pool of adaptive alleles, including at enhancers linked to inflammatory diseases. We propose that primate-specific transposons facilitated the evolution of inflammatory responses in great apes, with Alus shaping adaptive potential in modern humans.

Inflammatory immune responses are vital for survival, acting as a defense mechanism against various threats. However, inflammation is also involved in nearly all modern diseases[1,2]. The genetic roots of this paradox are embedded in human evolutionary history. For instance, mutations selected to bolster immune defense in past environments can, under modern conditions, heighten the risk of chronic inflammation[3–6]. Evolutionary events predating human history may have laid a more ancient foundation for contemporary inflammatory responses. A recent study shows that great apes, including humans, exhibit a more robust and broader early transcriptional response to immune stimulation compared to monkeys[7]. This response is enriched in inflammatory pathways[7], suggesting rapid regulatory evolution of inflammation in hominids. While this adaptation may protect population fitness in primates with increased body size and delayed reproductive maturity, it may also predispose them to chronic inflammatory diseases.

Inflammatory responses are under the control of transcriptional enhancers that often harbor risk alleles and are recognized as crucial elements of evolutionary adaptability[8]. Enhancers evolve through mutations and expansion of DNA sequences, altering transcription factor binding sites (TFBS) and, thereby, gene expression[9–12]. Throughout evolution, transposable elements (TEs) have been a major source of novel DNA[10,13], contributing not only raw sequence material but also regulatory motifs that facilitate enhancer evolution[14]. Although most TEs are dormant today, they once spread to occupy nearly half of the mammalian genomes[15]. Based on their transposition mechanism, either through DNA or RNA intermediates, TEs are classified into DNA transposons and the more prevalent retrotransposons, which include endogenous retroviruses (ERVs), long interspersed nuclear elements (LINEs), and short interspersed nuclear elements (SINEs, including Alus in primates)[16], all further branching into smaller groups and subfamilies[17].

[1]Institut Curie, PSL University, INSERM U932, Immunity and Cancer, Paris, France. [2]Institut Pasteur, Université Paris Cité, CNRS UMR2000, Human Evolutionary Genetics Unit, Paris, France. ✉e-mail: ella.zueva@curie.fr

Throughout primate evolution, periodic invasions by primate-specific TEs have profoundly reshaped genomes and regulatory networks[10,18–20]. The link between these TEs and immunity has been recognized previously. For example, retrovirus Mer41B, through interferon response motifs, contributes to the IFN-I responses associated with immunity and inflammation[21]. Alus are particularly biased towards immune-cell regulatory elements[22] and, along with other primate-specific TEs, play a significant role in the formation of novel enhancers[19]. Even in non-immune tissues, such as liver, active enhancers harboring primate-specific transposons are located near genes associated with immunity and inflammation[19].

Here, we employed comparative genomics and population genetics to systematically explore how primate-specific transposable elements (pTEs) have shaped the evolution of human immune-cell enhancers across distinct evolutionary time scales. By reconstructing the history of sequence divergence, pTE acquisition, and the emergence of proinflammatory TFBS following the human-macaque split, we show that pTEs introduced a substantial number of motifs for inflammation-related transcription factors. Integrating large-scale ChIP-seq datasets, we reveal that these pTE-derived motifs actively promote the in vivo binding of the corresponding proinflammatory NF-κB and IRF1 proteins. The significance of pTE-derived binding sites has grown over evolutionary time as they have become key contributors to functional great-ape-specific inflammation-related TFBS. Furthermore, NF-κB1 and IRF1 motifs derived from Alu elements and primate ERVs, respectively, exhibit signatures of fine-tuning in primates and humans, with a consistent trend toward increased predicted binding affinity. We emphasize the important role of Alus as highly mutable elements and hotspots of positive selection in humans, serving as a major source of NF-κB1 motifs under selection. Our findings highlight the profound impact of ancestral pTEs on the evolution of human inflammatory responses, revealing a vast potential for future adaptations.

## Results

### Primate-specific TEs are enriched in rapidly diverging immune-cell enhancers

To investigate the role of pTEs in the regulatory evolution of human inflammatory response, we first compiled a comprehensive list of putative human immune-cell enhancers. We retrieved annotations for various lymphoid and myeloid populations from Enhancer Atlas 2.0[23] and for lymphoblastoid cell lines from Garcia-Perez et al.[24]. Overlapping coordinates were merged to obtain a unified set (Supplementary Fig. 1a). To select robustly active elements, we integrated publicly available datasets of ATAC-sequencing from both unstimulated and stimulated immune cell populations[24–26] (Supplementary Data 1), yielding 60,332 regions. Most regions display broad accessibility across immune populations, with aggregated ATAC-seq signal delineating their boundaries (Supplementary Fig. 1b). Consequently, enhancers often contain multiple ATAC-seq peaks and cluster based on shared accessibility into groups common to all or several cell types, with only a small proportion being cell-type specific (Supplementary Fig. 1c, d). In most cases, the final size of the merged region matched the coordinates of the largest overlapping enhancers, indicating that we captured the broadest boundaries of regulatory hubs, which are frequently used by immune cells either in part or as a whole (Supplementary Fig. 1e, left panel). The final size distribution (median ~3 kb) resembled that of individual cell types, with no shift toward longer elements but a reduced representation of shorter ones, consistent with the consolidation of overlapping coordinates (Supplementary Fig. 1e, right panel). Given their promiscuous profile across immune cell types, we refer to these regions hereafter as immune-cell enhancers.

To trace the evolution of DNA sequence within these human immune-cell enhancers back to macaque, we converted their coordinates to high-quality genome assemblies of rhesus macaque (RheMac10) and chimpanzee (PanTro6), representing our closest monkey and great ape relatives, which diverged approximately 30–35 and 6–12 million years ago, respectively[27]. Since human and chimpanzee genomes differ by only a few percent[28], we defined orthologs as regions with a minimum of 97% alignability between species. Using the UCSC LiftOver converter, we classified enhancers into three groups based on their alignability under these conditions: (i) three-species orthologs (30,058), (ii) human/chimpanzee orthologs (25,479) carrying human-chimp sequence variations, and (iii) regions with human-specific variations (4795) (Fig. 1a and Supplementary Data 2). Based on the evolutionary timing of sequence divergence, these groups were defined as static, intermediate, and rapid, with the latter two collectively referred to as dynamic regions (Fig. 1a). A similar analysis using length-normalized (±1000 bp from midpoints) or individual cell type enhancers yielded overall comparable classification patterns, indicating that the merging procedure did not substantially bias the analysis (Supplementary Fig. 1f). While dynamic regions were not alignable to the macaque genome at a given threshold, lowering it to 50% alignability identified quasi-orthologs for 96% of them, indicating that the alteration of ancestral sequences is more prevalent than the emergence of enhancers from entirely novel DNA.

We then used PhastCons to measure sequence conservation across 17 primate clades, revealing that dynamic regions are significantly less conserved than static regions over this broader evolutionary scale (Fig. 1b). To estimate the rate of point mutations independently of indels, we analyzed gap-free BLASTn alignments between human and macaque within each enhancer group. Dynamic regions exhibited a significantly higher frequency of nucleotide mismatches per enhancer compared to the static regions (Fig. 1c). For all enhancer groups, most mismatches are not adjacent to alignment gaps (Supplementary Fig. 2a). Their density increases beyond ~50 bp from the nearest gap boundary, peaking between 100 and 300 bp, which rules out gap-related misalignment artifacts. Together, these results suggest that, compared to static regions, dynamic ones have evolved under lower genetic constraint, i.e., weaker purifying selection, which permits a higher TE frequency[29].

To measure pTE content in distinct enhancer groups, we annotated TEs via RepeatMasker (www.repeatmasker.org), distinguishing primate-specific biotypes using clade information from the TEanalysis tool (https://github.com/4ureliek/Teanalysis). To access the timing of pTE acquisition following the human-macaque split, we classified their sequences into two categories: (i) shared pTE sequences, inherited from the human-macaque common ancestor, as defined by conserved genomic location between human and macaque, and (ii) gained pTE sequences, acquired later in evolution and identified by any overlap with genomic gaps in macaque or chimpanzee relative to human. Of note, the gained pTEs could have originated from the reactivation of pre-inserted pTE species or the invasion of newly emerging ones. Although likely a minority, some evolutionary gains may have arisen from the loss of the original pTE sequences due to genome re-shuffling during speciation.

This analysis revealed that, while a large fraction of enhancers in all three groups overlap shared pTEs by at least 10 bp (Fig. 1d, left), dynamic regions are significantly more likely to do so than static ones (69% and 64% vs. 53%). They are also more likely to have gained pTEs since the split from macaque, with nearly half of the dynamic regions intersecting with sequences absent in macaque, compared to only 2% of the static enhancers (Fig. 1d, middle), and with rapid regions accounting for the majority of further human-specific gains (31%) (Fig. 1d, right). This trend was not driven by peripheral pTE accumulation, as it persists across central enhancer regions, as confirmed by length normalization to ±500 and ±1000 bp from the enhancer midpoint (Supplementary Fig. 2b). Dynamic regions, including size-normalized sets, exhibited a greater median coverage by pTE sequences compared to the static regions (~22% vs. ~16%; Fig. 1e, left;

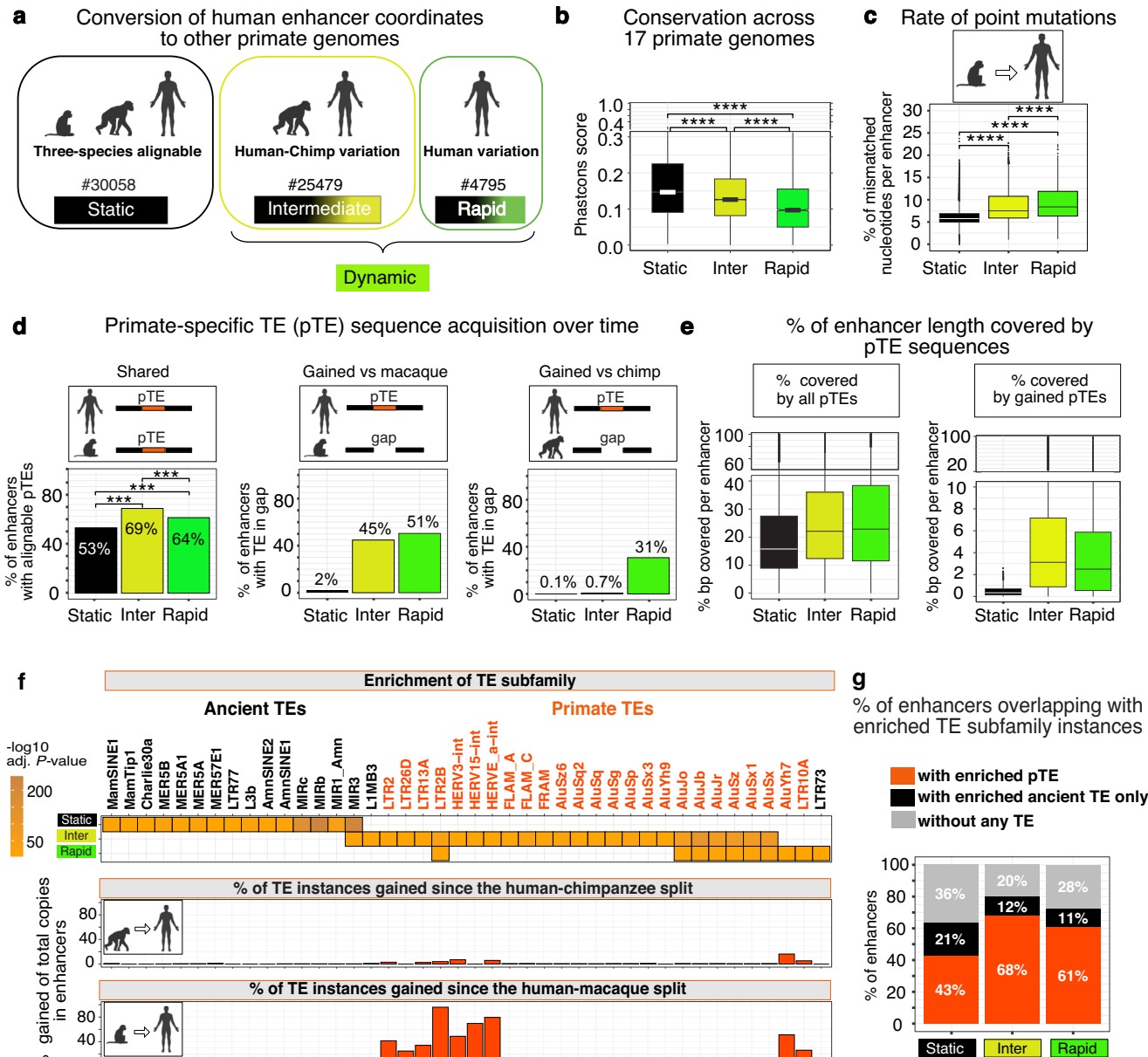

**Fig. 1 | Sequence evolution in putative human immune-cell enhancers since the human-macaque split. a** Stratification of human enhancers by evolutionary age based on their LiftOver conversion to the macaque and chimpanzee genomes. Number of enhancers in each group is indicated. **b** PhastCons conservation scores across 17 primate genomes. **c** Percentage of single-nucleotide substitutions per enhancer in human relative to macaque based on BLASTn alignment. In panels (**b, c**), comparisons were performed using pairwise two-sided $t$ test with $P$-values adjusted for multiple testing with the Benjamini–Hochberg method. **** $P \sim 0$. **d** Proportion of enhancers overlapping with pTEs, either present in the same genomic location in macaque (shared) or corresponding to genomic gaps (gained) in macaque (middle panel) or chimpanzee (right panel) compared to human. Pairwise two-sided χ² tests were applied with Benjamini–Hochberg correction ***$P < 0.001$. Exact values: $P = 1.12e\text{-}37$ (rapid vs inter), $P = 8.13e\text{-}28$ (rapid vs static), and $P \sim 0$ (inter vs static). **e** Fraction of enhancer length covered by pTE sequences,

considering either all pTE (left) or only those absent from macaque (right). **b, c, e** boxplots show the median (centre line), 25th-75th percentiles (box), and minima and maxima within 1.5× IQR (whiskers). **f** TE subfamilies most significantly enriched in enhancers, but not enriched in 1,000 times permutations matched for enhancer group number, length distribution, and LiftOver-based classification. Scale shows hypergeometric enrichment with Benjamini–Hochberg-adjusted $P$-values. Represented are pTEs with $P < 10e\text{-}10$, fold change ≥ 2, $n \geq 10$. Middle and bottom: proportion of the enhancer-linked pTE instances gained since the split from chimpanzee (middle panel) or macaque (lower panel). **g** Proportion of enhancers overlapping a pTE (≥10 bp), an ancient TE only (no pTE), or no TE. "Inter" represents intermediate enhancers throughout. The number of enhancers per group is given in panel **a** and applied throughout. Primate icons are created in BioRender. Zueva, E. (2025) https://BioRender.com/8d7b1bb. Source data are provided as a Source Data file.

Supplementary Fig. 2c). Notably, pTEs inserted after the human-macaque split accounted for a median of 2–3% of dynamic enhancer length, tenfold higher than in static regions (Fig. 1e, right panel). Together, these patterns correspond to the low genetic constraint identified above in dynamic regions, as this facilitated pTE accumulation.

To identify enriched TE subfamilies, we compared their abundance in distinct enhancer groups to a genome-wide background accounting for both copy number and genomic coverage (by length), using both hypergeometric and binomial statistical tests (adjusted $P$-value < 0.01). To control for biases from enhancer size and the Lift-Over, we shuffled 1000 times random non-enhancer regions matching

immune-cell enhancers in terms of number and length distribution. Each shuffled set was split into three groups: (i) by mirroring individual enhancer groups in number and length distribution, and (ii) using the same LiftOver-based classification as above, with TE enrichment assessed in each. A TE subfamily was defined as enhancer-enriched if significantly overrepresented in enhancer groups versus the genome by both hypergeometric and binomial tests (by counts and coverage) and not enriched by either test in matched random regions from (i) and (ii) (Supplementary Data 3). We observed enhancer groups exhibiting distinct patterns, with ancient TE subfamilies enriched in static regions and pTE subfamilies, mostly Alus and ERVs, predominantly enriched in the dynamic regions (Fig. 1f, upper panel). This trend persisted when enhancers were length normalized to ±500 bp and ±1000 bp from their midpoints (Supplementary Fig. 2d).

To trace the acquisition of these enriched TEs over evolutionary time, we mapped their coordinates in human enhancers to the corresponding genomic gaps in macaque and chimpanzee, requiring a minimum 90% overlap. As expected, we observed no gain of ancient TE copies since the human-macaque split, as their genomic loci are mainly conserved (Fig. 1f, middle and lower panels). For pTEs, acquisition timing mirrors evolutionary age, with the youngest ERV species mostly gained after the human-chimpanzee split. Additionally, 10–15% of older AluJ elements, which colonized early primate genomes, and intermediate-aged AluS elements, which emerged in monkeys[30], were gained after the human-macaque split, suggesting their reactivation in the common human-chimpanzee ancestor (Fig. 1f, middle and lower panels). Altogether, pTE instances from enriched subfamilies overlap (by at least 10 bp) with 68% of dynamic enhancers, compared to 43% of static enhancers (Fig. 1g).

Notably, Alu elements, 300 base pair long retrotransposons derived from the 7SL gene[31], are the most abundant pTEs in the human genome[30]. In immune-cell enhancers, they account for nearly 99% of all instances from enriched pTE subfamilies, with the second most abundant biotype (1%) being primate-specific ERVs (pERVs) (Supplementary Fig. 3a). Functionally, Alus can carry enhancer-specific histone marks at nucleosomes[32], while ERVs are known to contribute to accessible chromatin[20], characterized by transcription factor binding hotspots. To examine how these elements contribute to chromatin accessibility, we first quantified the enrichment of TE subfamilies at enhancer-overlapping immune-cell ATAC-seq peak summits. ERVs were significantly overrepresented across all enhancer groups, with dynamic regions containing a greater diversity of pERV subfamilies (Supplementary Fig. 3b, top panel). At the same time, Alu subfamilies showed stronger enrichment at the summits of dynamic regions compared to those of static regions (Supplementary Fig. 3b, bottom panel). To quantify their contribution, we classified ATAC-seq peak sequences as shared DNA if ≥50 aligned to the macaque genome and novel DNA if >50% overlapped a genomic gap in the macaque. Alu- or pERV-derived peaks were defined based on a ≥50% overlap with Alu or pERV sequences. In these peaks, the highest ATAC-seq signal intensity, potentially marking the origins of chromatin accessibility, aligns with pTE elements (Supplementary Fig. 3c). Within shared DNA, pERVs and Alus contribute up to 4% and 2% of ATAC-seq peaks, respectively (Supplementary Fig. 3d). Their contribution increases sharply in evolutionary novel DNA, with pERVs and Alus each generating a significant proportion of open chromatin, accounting for nearly 16% and 20%, respectively (Supplementary Fig. 3d). These findings show that Alus and pERVs together contribute substantially to accessible chromatin in immune-cell enhancers, particularly within DNA gained since the human-macaque split.

In summary, we identified human immune-cell enhancers that were particularly prone to accumulating pTE sequences after the human-macaque split, likely due to low genetic constraint. While pTEs contributed to regulatory hotspots of accessible chromatin within enhancers, their impact was especially pronounced during the latest stages of primate evolution.

## Dynamic regions are primarily enriched for inflammation-related TFBS

To investigate the evolution of inflammation-related transcription factor (TF) binding sites across enhancer groups, we systematically scanned these regions for overrepresented TF binding sites (TFBS) using the HOMER collection of ChIP-seq-instructed motifs. Compared to the genomic background, static regions were enriched for a wide array of TFBS, including for the inflammation-related IRF family proteins, which mediate interferon I response during viral infections and inflammation[33], and NF-κB, critical for initiating and resolving inflammatory responses[34] (Fig. 2a). Notably, NF-κB is a multiprotein family including NF-κB1, NF-κB2, RELA, RELB, and RELC subunits, all of which bind to variations of the same consensus sequence[34]. Additional enriched TFBS in static regions included those for NF-E2-related factors, antagonists of NF-κB[35], and JUN, ETS, and RUNX family proteins, which play diverse roles in immunity[36,37]. Enrichments were also observed for zinc finger proteins and factors that influence the identity of immune cells. In contrast, dynamic regions were enriched for a more limited set of predominantly inflammation-related TFBS (Fig. 2a). Moreover, corresponding motifs from the JASPAR2022 database are also enriched in enhancer-associated immune-cell ATAC-seq peaks (Supplementary Data 4) with a substantial fraction, ranging from 24% to over 50%, fully contained within peak boundaries (Fig. 2b), supporting their potential functionality.

To trace the evolutionary gain of potential binding sites in enhancers since the human-macaque split, we focused on TF families enriched in enhancer groups. Human JASPAR2022 UCSC coordinates of their respective motifs were intersected with the genomic gaps in macaque, chimpanzee, and gibbon (an earlier-diverging lesser ape) relative to human. TFBS fully overlapping these gaps (by 100%) were classified as gained. Quantifying the proportion of gained TFBS relative to the total number of enhancer-associated sites of the same type revealed that 6–11% (depending on TFBS class) are absent from both macaque and gibbon genomes (Fig. 2c, left and middle) but broadly shared with chimpanzee, with only ~1% being human-specific (Fig. 2c, right). This implies that the major TFBS expansion occurred in the human–chimpanzee common ancestor. Therefore, we classified the gained TFBS as great-ape-specific. In enhancers, they account for thousands of potential binding sites for each TFBS class (Supplementary Data 4). Among these, motifs for proinflammatory factors IRF1 and NF-κB1 showed the most rapid gains, as evidenced by the highest proportion of gained relative to their respective total pools in enhancers (Fig. 2c).

We then focused on inflammation-related TFBS, including a set of motifs for the major members of NF-κB family. Dynamic regions were significantly more enriched for NF-κB1, NF-κB2, RELA, and IRF1 motifs compared to the static regions used as background (adjusted P-value < 1e-10) (Fig. 2d). Moreover, inflammation-related TFBS were gained almost exclusively within dynamic enhancers (Fig. 2e), with a large proportion of these motifs fully embedded within immune-cell ATAC-seq peaks (Fig. 2f). These findings point to a distinct role for dynamic enhancers in evolutionary adaptation.

To access the potential functional role of dynamic regions, we linked individual enhancers to genes associated with immune response (from Hawash et al.[7]) using activity-by-contact (ABC)-predicted enhancer-gene interactions across immune populations[38,39] (Supplementary Data 5). The ABC model improves enhancer-gene assignment by integrating enhancer activity, 3D contact frequency, and functional impact on gene expression, as validated by CRISPR interference[39]. We observed a significant overlap in gene targets between distinct enhancer groups, implying regulation of the same functional pathways (Supplementary Fig. 4a). However, immune-response genes are associated with a higher number of intermediate enhancers (median of five) compared to other enhancer categories (Supplementary Fig. 4b). In contrast, rapid regions are less broadly

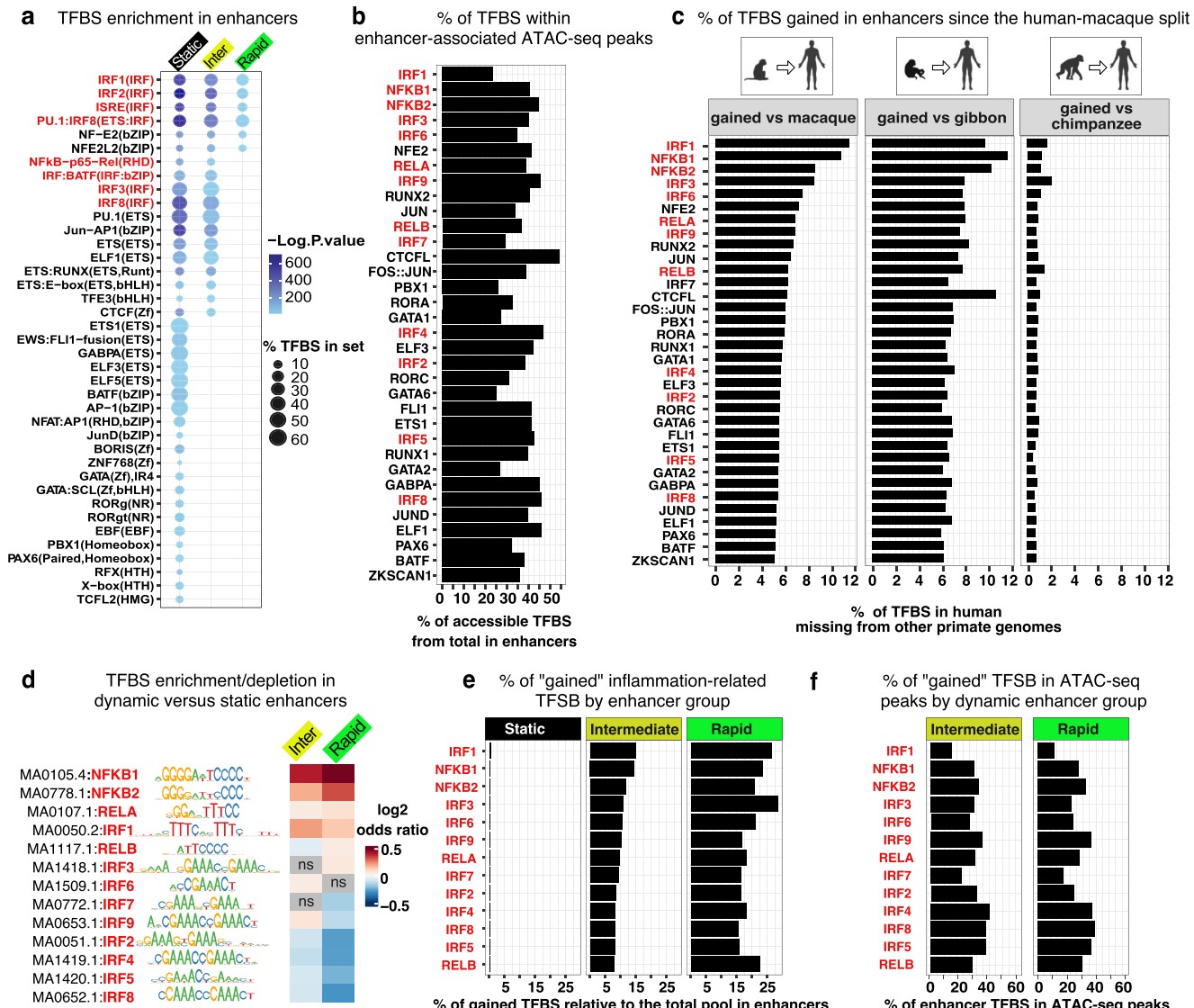

**Fig. 2 | Evolution of the enhancer regulatory lexicon. a** Transcription factor binding sites (TFBS) from the HOMER ChIP−seq motif collection significantly enriched in distinct enhancer groups, as assessed by HOMER's hypergeometric motif enrichment test. Shown are TFBS with $P < 0.01$, $\geq 5\%$ sequences with motif, and fold change $\geq 10$. Inflammation-related TFBS are highlighted in red throughout. **b** Proportion of enhancer TFBS fully embedded in the enhancer-localized immune-cell ATAC-seq peaks. **c** Proportion of enhancer TFBS mapping to genomic gaps in macaque (RheMac10), gibbon (NomLeu3), or chimpanzee (PanTro6), relative to the total number of corresponding motifs in enhancers. **d** Enrichment/depletion of NF-κB and IRF family motifs in intermediate and rapid enhancers relative to static assessed by two-sided $\chi^2$ test. **e** Proportion of TFBS matching macaque genomic gaps relative to total TFBS in each enhancer group. **f** Proportion of enhancer TFBS fully embedded in the immune-cell ATAC-seq peaks located within dynamic enhancer groups. "Inter" represents intermediate enhancers throughout. Primate icons are created in BioRender. Zueva, E. (2025) https://BioRender.com/8d7b1bb. Source data are provided as a Source Data file.

engaged, likely due to their low number and recent evolutionary footprint. We then split genes into two comparably sized sets with an intermediate-to-static enhancer ratio above or below one (Supplementary Fig. 4c). Genes biased towards intermediate enhancers (higher ratio) were more enriched for immune-related terms, including inflammation, based on both $P$-value and gene count per term compared to genes biased towards static enhancers (lower ratio) (Supplementary Fig. 4d). These findings suggest that intermediate enhancers may provide greater regulatory redundancy in immune-response regulation than other enhancer categories.

In conclusion, inflammation-related TFBS present within human immune-cell enhancers but absent in macaque were primarily gained during the evolution of the human-chimpanzee common ancestor. These TFBS predominantly emerged within dynamic regions, where IRF1 and NF-κB motifs expanded more rapidly than other types of TFBS. Our findings raise the possibility that

intermediate enhancers contribute to the regulatory adaptability of the immune response while static enhancers support its conserved core functions.

## pTEs disseminated inflammation-related TFBS, reshaping transcription factor binding in immune-cell enhancers

To assess the role of pTEs in inserting TF binding sites, we overlapped selected TFBS coordinates from the human JASPAR2022 UCSC tracks with locations of enhancer-associated pTEs. We retained the previous classification of TFBS as shared (with conserved human-macaque locations) and great-ape-specific (fully contained within genomic gaps in macaque). The analysis revealed that pTEs made significant contributions to shared TFBS, accounting for 10% to 50% of the sites, depending on the TF class (Fig. 3a, left). However, their contribution to great-ape-specific TFBS is markedly higher, with pTEs in some cases serving as a critical source, particularly for NF-κB-related transcription

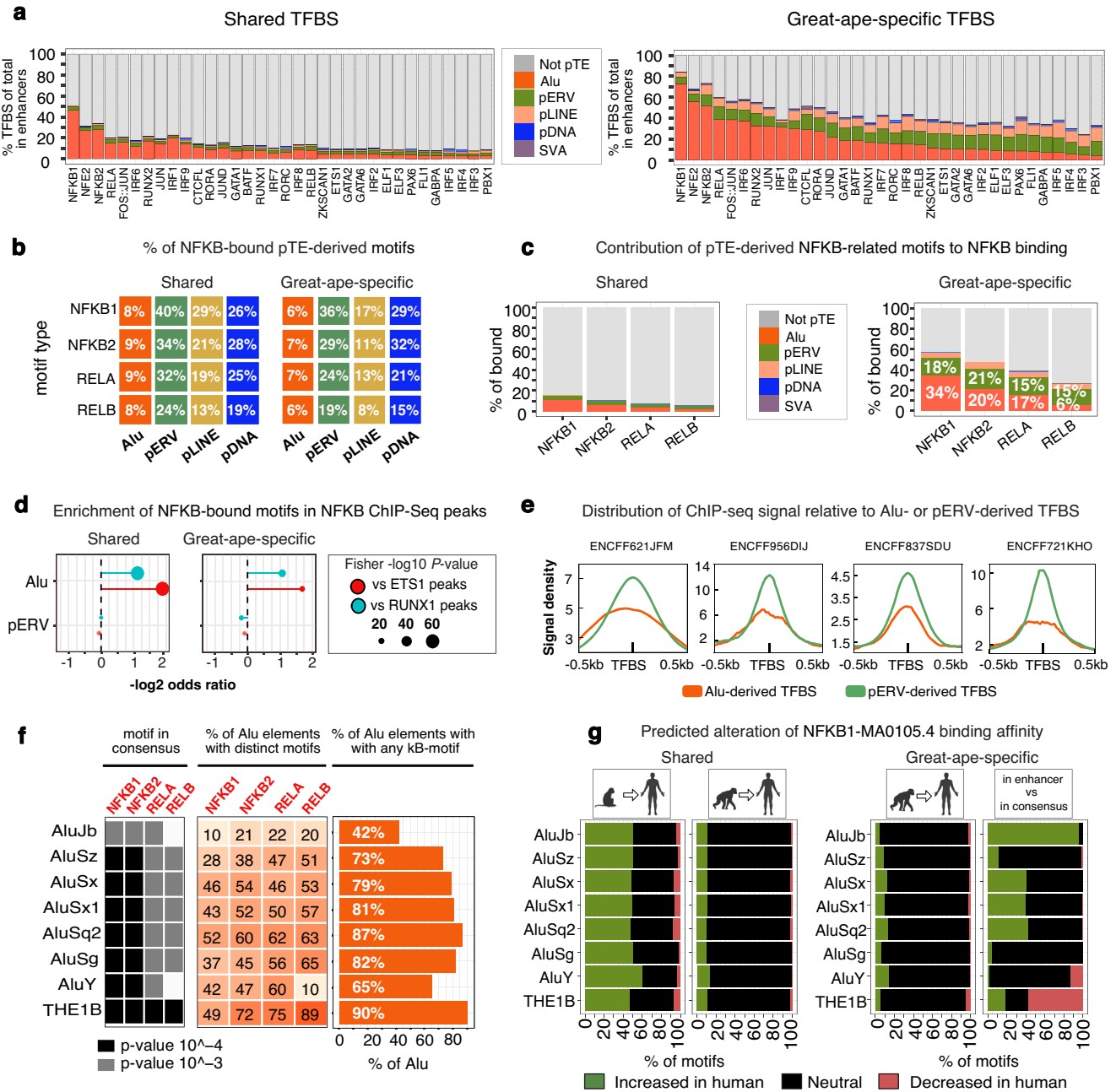

**Fig. 3 | Evolution and functionality of the pTE-derived enhancer-associated NF-κB motifs. a** Proportional contribution of sequence types to TFBS in enhancers, with pTE-derived TFBS highlighted in color. Left: TFBS shared with macaque. Right: great-ape-specific TFBS (fully contained within genomic gaps in RheMac10 relative to GRCh38). **b** Fraction of NF-κB-bound TFBS (fully embedded within ChIP-seq peaks) within each pTE biotype. **c** Contribution of enhancer sequence categories to NF-κB-bound motifs; pTEs highlighted. **d** Enrichment/depletion of Alu- or pERV-derived NF-κB motifs embedded within NF-κB ChIP-seq peaks, relative to the same motifs within ETS1 and RUNX1 ChIP-seq peaks assessed using two-sided Fisher's exact test. **e** Distribution of NF-κB ChIP-seq signal centered on Alu and pERV-derived motifs. **f** Left: NF-κB motifs identified in consensus sequences of pTE

subfamilies most abundant within NF-κB ChIP-seq peaks (>100 copies). *P*-values are computed by FIMO (MEME suite) with Benjamini-Hochberg correction; *P* < 0.0001 is considered a stringent match (black) and *P* < 0.001 a permissive match (gray). Middle: fraction of enhancer-localized pTEs carrying each distinct NF-κB motif. Right: Fraction of pTEs in enhancers carrying any of the four NF-κB motifs. **g** Left panels: predicted binding affinity differences (Δ) for shared pTE-derived NF-κB1 motifs (MA00105.4) between human and macaque/chimpanzee genomes (TFBStools). Right panels: Δ for great-ape-specific motifs relative to chimpanzee orthologs and consensus sequences. Inflammation-related motifs are highlighted in red throughout. Primate icons are created in BioRender. Zueva, E. (2025) https://BioRender.com/8d7b1bb. Source data are provided as a Source Data file.

factors (Fig. 3a, right). For example, Alu elements contribute over 70% of great-ape-specific NF-κB1 motifs and more than 40% of NF-κB2 motifs. Most of these Alu elements (90%) precisely match genomic gaps in macaque (≥ 97% overlap), suggesting that new Alu insertions were the primary drivers of NF-κB motif expansion in the human-chimpanzee common ancestor. Noteworthy, Alus are not the most frequent sequences that underlie TFBS-corresponding gaps. Most

other great-ape-specific TFBS originate from non-Alu DNA (Fig. 3a, right).

We then focused on NF-κB-related motifs, which represent some of the most rapidly expanded sites since the human-macaque split and potentially bind key regulators of inflammation. To evaluate the in vivo functionality of pTE-derived NF-κB-related motifs, we analyzed ChIP-seq data from the Remap2022 and ENCODE consortia, covering

various tissues (Supplementary Data 6). Given that NF-κB factors can bind different NF-κB-related motifs through complex and not fully understood interactions[40,41], we combined ChIP-seq peaks for NF-κB1, NF-κB2, RELA, and RELB proteins. Binding capacity was assessed for a set of NF-κB-related motifs from the JASPAR2022 collection, all of which are enriched within ChIP-seq peaks (Supplementary Data 4). For each pTE biotype, we quantified the proportion of bound motifs, defined as TFBS fully embedded (100% overlap) within ChIP-seq peaks whose summits intersect immune-cell enhancers. pERVs exhibited the highest proportions of motifs capable of binding, with 40% of shared pERV-derived NF-κB1 motifs found within ChIP-Seq peaks (Fig. 3b, left). In contrast, a smaller fraction of Alu-derived NF-κB-related motifs were bound (8% of shared NF-κB1 motifs). This pattern is consistent across both shared and great-ape-specific motifs (Fig. 3b, left and right).

Next, we quantified the proportion of pTE-derived among all bound NF-κB-bound motifs within enhancers. Overall, the proportion of pTE-derived motifs was remarkably higher among great-ape-specific compared to shared bound motifs (Fig. 3c). For instance, over 50% of bound great-ape-specific NF-κB1 motifs are pTE-derived, with the majority originating from Alu sequences (Fig. 3c, right). These results indicate that pTE-derived TFBS contribute substantially to inflammation-related TF binding, with great-ape-specific motifs showing intrinsically higher binding propensity.

Given that Alus and pERVs contribute more NF-κB-bound sites compared to other pTE biotypes (Fig. 3c and Supplementary Fig. 5a), we further focused on these two pTE groups. Alu-derived motifs were generally less likely to be bound relative to their overall abundance in enhancers (e.g., OR = 0.18, $P = 2.2e−16$ for shared NF-κB1 motifs in NF-κB peaks). In contrast, pERV-derived motifs were enriched in ChIP-seq peaks relative to their frequency in enhancers (e.g., OR = 1.2, $P = 7e−05$ for shared NF-κB1 motifs). Yet, despite having a lower binding like-lihood compared to pERV-derived NF-κB-related motifs, Alu-derived ones remained more prevalent within ChIP-seq peaks by the absolute number (Supplementary Fig. 5a) due to their abundance.

Given the possibility of accidental overlap with ChIP-Seq peaks, we assessed the specificity of the observed binding by comparing the enrichment of Alu- and pERV-derived NF-κB motifs within NF-κB ChIP-seq peaks against those of unrelated proteins, such as ETS1 and RUNX1. The analysis revealed significant enrichment of both shared and great-ape-specific Alu-derived but not pERV-derived NF-κB-related motifs within NF-κB peaks (Fig. 3d), highlighting Alu bias toward NF-κB activity. Altogether, Alu elements overlap (by at least 10 bp) with 4−20% of Chip-seq peaks, depending on the specific NF-κB protein type (Supplementary Fig. 5b).

We next analyzed ChIP-seq signal distribution relative to Alu- and pERV-derived motifs embedded within ChIP-Seq peaks. Although NF-κB signal is more diffusely distributed across Alu-derived motifs than pERV-derived ones, in both cases, elevated signal intensity aligns well with the motif position (Fig. 3e). A distinct pattern displayed by Alus may reflect a subset of motifs functioning in NF-κB trapping, consistent with the previously reported ability of nucleosomal DNA to prime NF-κB for binding[42,43].

Given the known contribution of ERVs to IRF1 binding[21], we performed a similar analysis using IRF1 ChIP-Seq data from the ENCODE consortium. pERV-derived IRF1 motifs were enriched in IRF1 peaks, whereas Alu-derived motifs were depleted (Supplementary Fig. 6a), suggesting pERV bias toward the IRF1 response. Approximately 24% of pERV-derived IRF1 motifs in enhancers were bound by IRF1 (Supplementary Fig. 6b). While pERVs contributed only a small fraction of shared motifs embedded within ChIP-seq peaks, their contribution increased markedly at great-ape-specific binding sites, reaching 14% (Supplementary Fig. 6c), with ~4% of enhancer-associated IRF1 peaks overlapping pERV elements (by at least 10 bp) (Supplementary Fig. 6d).

To assess whether NF-κB-related motifs were present as ready-to-use in the ancestral pTE sequences or arose through post-insertion

mutations, we focused on Alu and ERV subfamilies that contributed the most NF-κB-bound motifs in ChIP-Seq peaks. NF-κB-related motifs were searched for in subfamily consensus sequences from Dfam[44] by scanning JASPAR2022 position weight matrices (PWMs) using the FIMO tool. Motifs matching both $q$-value < 0.0001 (stringent) and $q$-value < 0.001 (permissive) were retained. NF-κB1 and NF-κB2 motifs were identified with high confidence ($q$-value < 0.0001) in most Alu subfamilies. In contrast, RELA/RELB motifs were identified with lower confidence ($q$-value < 0.001), indicating proto-motifs slightly deviating from their canonical PWM profiles (Fig. 3f, left panel). At the same time, all κB motifs were confidently detected in THE1B, the top NF-κB-binding ERV subfamily, previously associated with NF-κB binding[45,46] and antiviral immunity[47]. Although not all enhancer-associated copies retained the initial high-confidence motif (Fig. 3f, middle), most instances carried at least one canonical NF-κB motif type (Fig. 3f, right), indicating evolutionary transitions among related motifs.

A similar analysis of IRF1 motifs in IRF1 ChIP-seq peaks revealed that nearly all high-contributing ERV subfamilies harbored high-confidence IRF1 motifs in their consensus sequences, often pre-served in enhancer-associated instances (Supplementary Fig. 6e). Together, this suggests that functional motifs pre-existed in ancestral pTE sequences, rendering them potentially immediately available for regulatory co-option.

While ChIP-seq data cannot capture all inflammatory triggers, evolutionary shifts in TFBS binding affinity to cognate TF within enhancers may reflect past selective pressures and, potentially, func-tional relevance. Using the SearchSeq function in TFBStools[48], we predicted changes in binding affinity for the rapidly expanding Alu- and pERV-derived NF-κB1 motif (NF-κB1-MA0105.4) following the human-macaque split. For shared motifs, human sequences were compared to those of macaque and chimpanzee. For great-ape-specific TFBS, comparisons were made between human and chim-panzee and between enhancer and subfamily consensus sequences. Nearly half of the shared Alu-derived NF-κB1 motifs exhibited higher predicted binding affinity compared to macaque sequences, while a smaller proportion (10%) showed further increases relative to chim-panzee (Fig. 3g, left panels). This suggests that the main affinity shift occurred during the evolution of the common human-chimpanzee ancestor. Great-ape-specific NF-κB1 motifs showed minimal differ-ences between human and chimpanzee; yet, pTE-derived motifs in enhancers display both increased and decreased (in AluY and THE1B) NF-κB1 affinity compared to consensus sequences from Dfam (Fig. 3g, right panels), suggesting further optimizations. Given that AluS elements are more numerous than AluY or THE1B, the human evolutionary trend appears to favor an overall increase in binding affinity. Similar patterns were observed for the enhancer-associated pERV-derived IRF1 motifs (Supplementary Fig. 6f), as well as motifs embedded within ChIP-Seq peaks (Supplementary Fig. 5c and 6f), suggesting that beyond currently bound sites, enhancers contain numerous motifs undergoing binding affinity optimization.

In summary, pTEs have propagated proinflammatory NF-κB and IRF1 motifs across immune-cell enhancers, becoming a particularly rich source of the great-ape-specific binding sites. These motifs increase the activity of enhancers to bind their cognate transcription factors in vivo and tend to evolve toward higher binding affinity over evolutionary time.

## Alus are fertile substrates of positive selection in human

To investigate the impact of natural selection on pTEs and their associated enhancers in modern humans, we analyzed three major continental populations from the 1000 Genomes Project (Central Europeans from Utah (CEU), Yoruba from Nigeria (YRI), and Southern Han Chinese (CHS). We assessed positive selection signals in enhancer-overlapping single-nucleotide polymorphisms (SNPs) by integrating measures of genetic differentiation between populations (population

branch statistics)[49] with haplotype-based evidence of a rapid increase in derived allele frequency, as provided by Relate[50]. This analysis identified 50,000 enhancer-associated SNPs with signatures of positive selection (PS-SNPs, Fisher's combined $P < 0.01$; Supplementary Data 7). We further identified positive selection acting on entire enhancer loci by taking the strongest selection $P$-values across enhancer-overlapping SNPs and adjusting for multiple testing to obtain a single selection $P$-value per enhancer. Applying a 5% false discovery rate (FDR) to these $P$-values, we identified 3,500 enhancers that present signatures of positive selection of the whole region (PS-enhancers; Supplementary Data 7).

We first examined the correlation between pTE presence and positive selection at the enhancer level. Our analysis revealed that enhancers containing Alu elements are significantly more likely to be under positive selection in modern humans than those without, particularly when the Alu sequences were acquired after the split from macaque (de novo Alus) (Fig. 4a, left panel). In contrast, pERVs and pLINEs do not correlate with positive selection, indicating this effect is Alu-specific. This pattern persists when enhancers are expanded ±5 kb from their midpoints to normalize by size and include nearby background (Supplementary Fig. 7a). The presence of NF-κB1 motifs further enhances the likelihood of positive selection, especially if these motifs are Alu-derived (Fig. 4b). This shows that Alu-derived NF-κB1 motifs alone can reflect specific dynamic properties of enhancers in modern human populations.

Next, we examined how pTEs themselves are targeted by positive selection within enhancers. By annotating enhancer sequences carrying SNPs, we observed that distinct pTE groups, ancient TEs, and non-TE sequences are similarly represented among both PS-SNPs and neutral SNPs (Fig. 4c, left and middle panels). This suggests that no specific type of enhancer sequence is favored by positive selection. Instead, Alus emerge as the most abundant pTE target, accounting for 15% of PS-SNPs, mainly due to their high frequency. Moreover, the proportion of Alu-derived SNPs is significantly higher than expected based on the overall Alu abundance (15% with PS-SNPs or neutral SNPs compared to 12% of Alu-derived enhancer sequences) (Fig. 4c). This implies that Alus are the most mutable elements in enhancers.

To determine whether selection acted directly on pTE-derived NF-κB motifs, we focused on enhancer-linked PS-SNPs overlapping NF-κB motifs. By analyzing the contribution of different enhancer sequence types to NF-κB motifs under positive selection, we found that NF-κB1-MA0105.4 is more frequently associated with Alu elements (60%) than with other sequences (Fig. 4d). Alus also significantly contribute to other related NF-κB motifs under positive selection, whereas other pTEs play a minimal role. At the same time, both PS-SNPs and neutral SNPs occur within Alu-derived NF-κB1 with comparable frequencies, with only a slight increase observed for the most significant PS-SNPs (Supplementary Fig. 7b). Furthermore, the overall proportion of SNPs under positive selection within NF-κB1 motifs is similar for Alu-derived and non-Alu-derived sequences (~6%, Supplementary Fig. 7c). Together, this suggests that Alus likely dominate the selection of NF-κB1-MA0105.4 due to being their richest source.

We found that SNPs modify the binding affinity of NF-κB1 motifs, whether they originate from Alu elements or less common sources such as ancient TEs or non-TE sequences (Fig. 4e). However, PS-SNPs exhibit a distinct pattern compared to neutral SNPs. In Alus, 30% of PS-SNPs increase the binding affinity of NF-κB1 motifs relative to the ancestral allele, compared to only 18% for neutral SNPs (Fig. 4e, left panel). Similarly, other positively selected NF-κB motifs show a marked shift toward higher binding affinity compared to neutral SNPs (Supplementary Fig. 7d), suggesting that recent positive selection has preferentially favoured alleles that increase the affinity of immune enhancers toward NF-κB.

We report two variants with strong selection signals within Alu elements in the PS-enhancers belonging to the intermediate group.

The first, rs9592968-A, in the AluSx1-derived NF-κB1 motif, which is shared with macaque (Fig. 4f), shows strong selection outside Africa (P-value_comb. <1e−7, P-value_relate <3e−6, P-value_PBS <3e−3 for CHS) and increases the binding affinity of this motif (Δ+8.94) relative to the ancestral G allele. The second, rs10818794-T (P-value_comb <1e−4, P-value_relate <7e−4, P-value_PBS <4e−3 CEU), reaching highest frequencies in Europe and the Indian Peninsula, resides in the enhancer mainly composed of novel DNA missing from the macaque genome and alters binding affinities of the underlying NF-κB2 and RELA motifs within great-ape-specific AluY (Δ−2.5 and +2.5, respectively) (Fig. 4g). Both rs9592968-A and rs10818794-T are significantly associated with altered white blood cell counts (Fig. 4f and g, lower panels), potentially linking them to inflammatory and autoimmune conditions[51].

Our results show that NF-κB binding sites introduced by Alu during primate evolution have served as fertile substrates for adaptation in modern humans. Recent positive selection has modulated the binding affinity of Alu-derived NF-κB1 motifs, likely contributing to the variation in susceptibility to inflammatory disorders among contemporary humans.

## Inflammatory disease-associated enhancers are the most adaptive in humans

To investigate how selection acts upon enhancers associated with inflammatory diseases, which are becoming increasingly prevalent, we constructed a core disease network. We first selected genes annotated in DisGeNet7.0 as associated with major inflammatory and autoimmune diseases and shared across at least one-third of these conditions (Fig. 5a). These annotations integrate curated databases and literature-mined evidence, capturing both well-established and emerging gene-disease links. Shared core disease genes are significantly enriched in the myeloid signature and targets for NF-κB1 and IRF1 transcription factors (Supplementary Fig. 8a, b). We then defined their putative enhancers based on (i) ABC-prediction and (ii) overlap with IRF1 or NF-κB ChIP-seq peaks. This yielded 9,767 regions potentially linked to inflammation, referred to hereafter as inflammatory disease enhancers or IDEs (Supplementary Data 8).

IDEs demonstrate unique patterns of evolution in primates, being more enriched in Alu elements than other enhancers (set as a background) (Fig. 5b, left panel) and more frequently harboring NF-κB1 motifs compared to non-IDEs (71% of IDEs vs. 49% of non-IDEs with motifs including 13% vs. 7% with great-ape-specific motifs, correspondingly) (Fig. 5b, right panels). Moreover, IDEs likely play an important role in the human adaptation of the inflammatory response, with 3% carrying SNPs reported as causative to chronic inflammatory and autoimmune disorders[8] compared to 0.7% of non-IDEs (Fig. 5c). A greater fraction of these regions contains PS-SNPs compared to non-IDEs (Fig. 5d, left), with a larger proportion of those PS-SNPs embedded within Alu elements (Fig. 5d, right). Together, these findings suggest that IDEs contributed to the adaptation of the inflammatory response throughout primate evolution, in part through the activity of Alu elements.

We examined the association of IDEs with genes that respond more strongly to immune stimulation in great apes compared to monkeys and in humans compared to other primates[7]. IDEs were strongly associated with both human-chimpanzee-specific and human-specific early transcriptional response genes (Supplementary Fig. 8c). For instance, over 40% of IDEs were linked to human-chimpanzee-specific genes, compared to just 6% of non-IDEs. This suggests a crucial role of IDEs in wiring this evolutionary novel immunity-related transcriptional response in the common human-chimpanzee ancestor.

A compelling example of the recent evolutionary adaptation of IDE is provided by the rs6011058-C/T variant occurring in the great-ape-specific AluYk2 and undergoing strong positive selection in Africa (P-value_comb <2e−5, P-value-relate <2e−5, P-value PBS <1e−2 for YRI) (Fig. 5e). The surrounding enhancer is primarily composed of

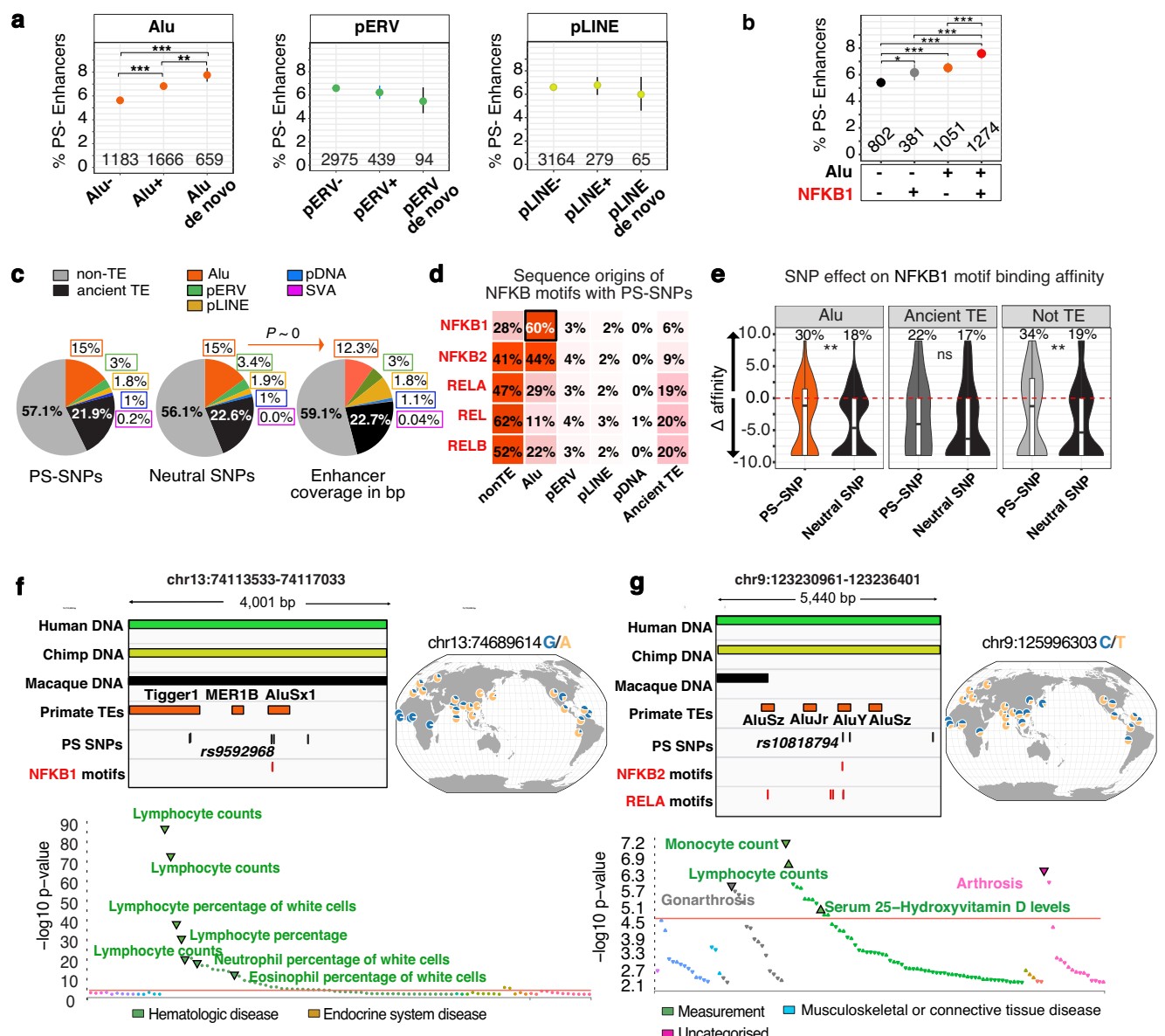

**Fig. 4 | Association of Alus with positive selection in humans. a** Proportion of PS enhancers in different groups (5% enhancer-level FDR) with numbers indicated. Exact values: $P = 5.25 \times 10^{-8}$ (Alu− vs. Alu +), $P = 4.57 \times 10^{-3}$ (Alu de novo vs. Alu +), and $P = 5.17 \times 10^{-10}$ (Alu de novo vs. Alu−). **b** Proportions of positively selected enhancers either with or without Alus, stratified by presence of Alu-derived or non-Alu NF-κB-MA0105.4 motifs. **a, b** Error bars represent 95% confidence intervals based on exact binomial tests. Statistical significance was assessed using pairwise two-sided Fisher's exact tests with Benjamini-Hochberg correction. ***$P < .001$, **$P < 0.01$, *$P < 0.05$. **c** Left and middle: proportions of sequence classes harboring PS-SNPs ($P < 0.01$) or neutral SNPs ($P > 0.5$) within enhancers. Right: proportion of total enhancer length occupied by each sequence class. pTE proportions are highlighted. Enrichment of Alus among SNP-containing sequences relative to Alu base-pair occupancy in enhancers assessed using a two-sided Fisher's exact test is indicated. $P \sim 0$. **d** Proportional distribution of NF-κB motifs with PS-SNPs, stratified

by their sequence origin. **e** Distribution of predicted NF-κB1-MA0105.4 binding affinity changes (Δ, derived vs. ancestral allele, estimated with TFBStools) for enhancer-localized PS-SNPs and neutral SNPs by sequence origin. Group differences in the proportion of motifs with positive Δ were evaluated with a two-sided $\chi^2$ test; exact values: $P = 0.0023$ (Alu), $P = 0.74$ (Ancient-TE), $P = 0.0026$ (not-TE). Boxplots show the median (centre line), interquartile range (box), and whiskers extending to 1.5× IQR. **f, g** Genome browser views of representative PS-SNPs located within Alu elements. Global selection patterns are visualized using the Geography of Genetic Variants Browser. PheWas analysis was performed using Open Targets Genetics (https://atlas.ctglab.nl/PheWAS) for traits significantly associated with rs9592968-A and rs10818794-T in FinnGen, UK Biobank, and GWAS Catalog. $P$-values are derived from GWAS regression models; the horizontal red line marks the Bonferroni significance threshold ($P < 1e-5$). Inflammation-related motifs are highlighted in red throughout. Source data are provided as a Source Data file.

great-ape-specific DNA enriched with Alu sequences and harbors an NF-κB1 motif 89 bp away from rs6011058. While the derived allele rs6011058-T does not alter an NF-κB1 binding affinity directly, it disrupts predicted binding sites for Krüppel-like factors KLF2, KLF3, and KLF15 (Δ $_{affinity}$ up to −9.0, Fig. 5e). KLFs are known to counteract NF-κB functions and reduce NF-κB-mediated transcriptional activity[52–54], thus preventing acute and chronic inflammatory conditions[54,55]. rs6011058-T has previously been identified as causal for Crohn's disease[8] and is

associated with other inflammatory disorders, such as atopic dermatitis and eczema (Fig. 5g). The disruption of KLF motifs may thus enhance the effect of NF-κB1 binding, increasing the activity of this IDE in driving expression of its target genes from the core inflammatory signature (RTEL1 and TNFRSF6B), as well as interferon-stimulated, great ape-specific immune response gene HELZ2[56].

In conclusion, we identified enhancers enriched in Alus and linked to genes frequently associated with inflammatory diseases.

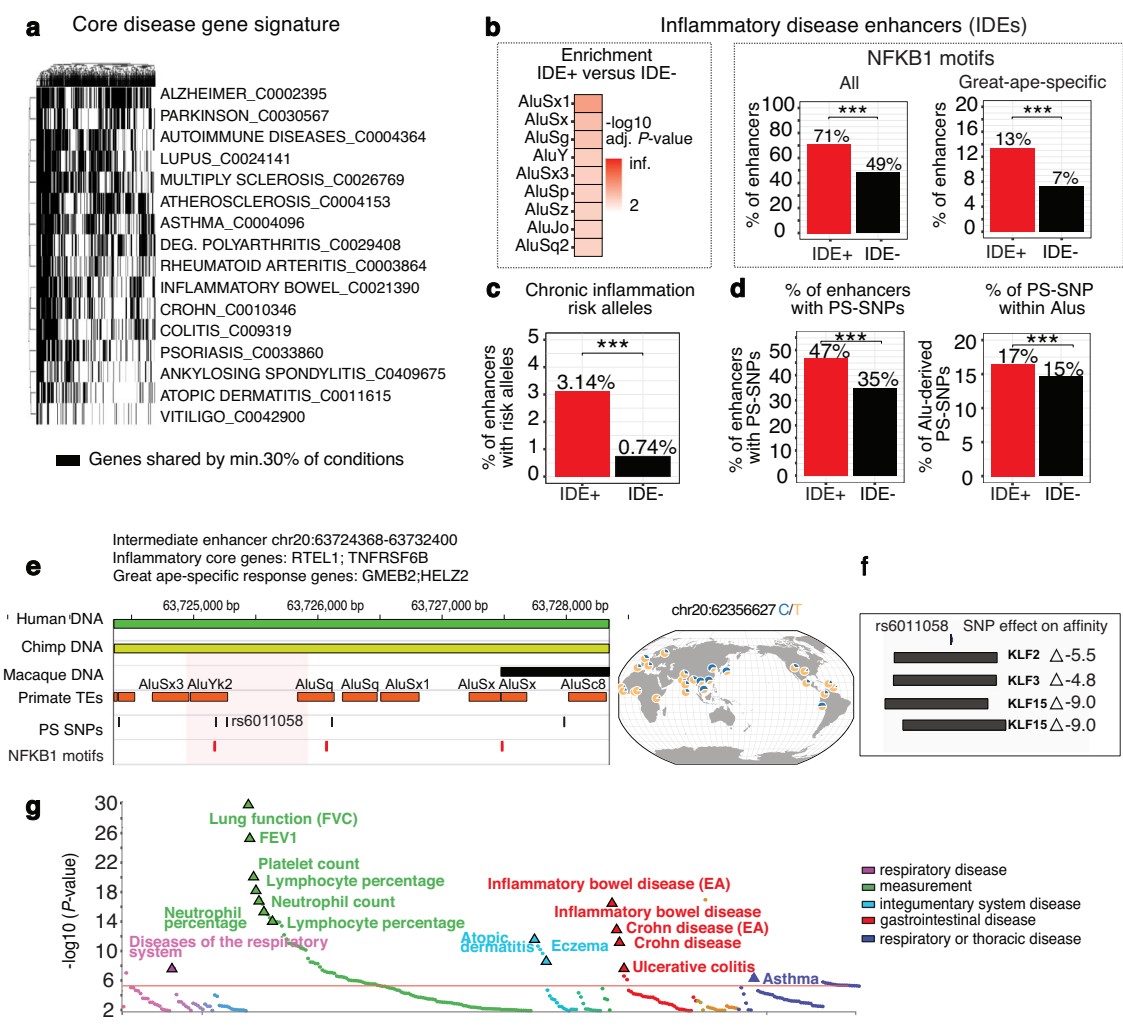

**Fig. 5 | Evolution of the inflammatory disease-associated enhancers. a** Binary map of genes (in black) shared among ≥ 30% of the major chronic inflammatory and autoimmune diseases, illustrating overlap between conditions. **b** Left: enrichment of Alus in IDEs relative to non-IDEs tested with a hypergeometric enrichment test and Benjamini–Hochberg correction (P < 0.1). Right: proportion of enhancers carrying NF-κB-MA0105.4 motifs; all (left) or great-ape-specific (right). **c** Proportion of IDEs and non-IDEs containing SNPs associated with chronic inflammatory and autoimmune disorders (P ~ 0). **d** Left: proportion of enhancers with PS-SNPs. Right: proportion of PS-SNPs within Alus relative to all PS-SNPs in enhancers in IDE versus non-IDE groups. Exact values: P ~ 0 (left), P = 2.03e-07 (right). **b** (right), **c, d** Statistical comparisons were performed using pairwise two-sided χ² tests. ***

P < 0.001. **e** Genome browser view of a fragment of positively selected IDE (from the intermediate group) carrying PS-SNP rs6011058-T (selection value P < 2e−5) within an AluYk2 element. Shown are genes linked to the enhancer via the ABC algorithm. Global selection patterns are visualized with the Geography of Genetic Variants Browser. **f** KLF motifs overlapping rs6011058-T and binding affinity changes of the derived versus ancestral allele, estimated with TFBStools. **g** PheWas plot was generated using Open Targets Genetics (https://atlas.ctglab.nl/PheWAS) for traits significantly associated with rs6011058-T in FinnGen, UK Biobank, and GWAS Catalog. P-values are derived from GWAS regression models; the horizontal red line marks the Bonferroni significance threshold (P < 1e−5). Source data are provided as a Source Data file.

These enhancers exhibit signatures of the inflammatory response adaptation in primates and humans, suggesting their potentially important role in the evolutionary shaping of the human inflammatory response.

## Discussion

Recent research has linked human inflammatory responses to adaptive genetic changes that occurred during human evolution. Here, we shift the perspective to a deeper evolutionary context, tracing the origins of human immune-cell enhancers to our primate ancestors, whose genomes were extensively restructured by the activity of young, primate-specific transposons. We demonstrate that, throughout evolution, these initially parasitic elements seeded enhancers with inflammation-related TFBS, which were subsequently repurposed for binding by their cognate transcription factors. The sharply increased regulatory contribution of pTEs during the evolution of the common human-chimpanzee ancestor likely reflects the rapid phenotypic changes in

large anthropoid primates. Such shifts may have required accelerated adaptation, with pTE-derived inflammation-related TFBS providing readily available, albeit potentially opportunistic, genetic tools to fine-tune inflammatory responses. Our findings suggest that pTEs not only shaped the interplay between immunity and environmental pressures in primates but also contributed rich substrates for natural selection and susceptibility to inflammatory diseases in humans.

More specifically, we observed a functional distinction between primate ERVs and Alus. While pERVs appear to be proficient binders of IRF1, as shown here and elsewhere[21], Alus are specifically biased toward NF-κB binding. Although their binding propensity in vivo is relatively low, Alu-derived NF-κB motifs may exert a greater influence on inflammatory responses than those from pERVs, owing to their substantially higher abundance in enhancers. The regulatory significance of Alus has progressively grown throughout evolution, as they became a key source of evolutionarily recent, great-ape-specific NF-κB1 motifs. Subsequently, Alus have served as rich substrates for positive selection

in humans, potentially contributing to the adaptation of NF-κB responses.

NF-κB motifs have previously been identified in Alu elements[57,58]. Here, we show that these motifs are embedded within ancestral Alus and appear to undergo continual optimization for improved TF binding within immune-cell enhancers. While not all Alu-derived NF-κB motifs are bound by their cognate proteins in vivo, this evolutionary fine-tuning may be explained by their utilization in response to threats that are not replicated in laboratory settings or by intermittent usage. It has indeed been shown that κB protein dimers may scan κB sites in a trial-and-error fashion to adjust transcriptional output[41,59]. In this scenario, prevalent Alu-derived NF-κB motifs may have been transiently sampled during proinflammatory stress events and positively selected if proven advantageous. This process may persist today, as the reservoir of unused motifs remains unsaturated yet available, given their presence in the favorable epigenetic environment of enhancers.

We found that Alus are associated with increased enhancer adaptability in humans and contribute disproportionately to positively selected sites within enhancers linked to chronic inflammatory diseases. This suggests that Alus play a role in facilitating responses to proinflammatory challenges. This ability may arise from the NF-κB1-MA0105.4 motif, which stands out as evolutionarily significant, having undergone the most rapid expansion in the common human-chimpanzee ancestor and now showing a strong association with adaptive enhancers in humans. Furthermore, the high mutability of Alu elements may expand the evolutionary potential of enhancers by increasing their likelihood of being targeted by natural selection. Finally, Alus are known to play an important role in enhancer-promoter looping by forming duplexes with complementary sequences[60] and to deliver transcription factors locally[58]. The iterative sampling of promoters by enhancer Alus, transiently bound by NF-κB, could further expand adaptive opportunities. The looping of three distantly located Alus to deliver NF-κB to the IFNb promoter during the early antiviral response, thereby jump-starting gene expression[58] may serve as an example of such a phenomenon in action. Although we do not explore the broader contribution of Alus to other TFBS, Alu-derived NF-κB antagonists such as NFE2 and KLFs may help to orchestrate a more complex regulation of the inflammatory response.

One might expect the abundance of Alus to dilute the impact of individual elements, with no single Alu being critical. However, the depletion of individual enhancer-residing Alu copies using CRISPR-Cas9 technology affects the expression of genes physically contacted by those enhancers, as shown in Liang et al. [60]. These findings suggest that Alu elements can act as minimal enhancers. While we did not perform Alu depletion experiments ourselves, we identified 38 Alu elements within immune-cell enhancers whose perturbation in Liang et al. was shown to impact gene expression (Supplementary Data 9).

We propose the concept of regulatory co-option of Alu elements by chance, driven by the law of large numbers. As Alus became prominent in dynamically evolving enhancers, they enabled rapid adaptation to environmental stimuli through pre-existing inflammation-related regulatory motifs. As such, Alu elements may not only reflect the survival history of our primate ancestors but also serve as a vast reservoir for future resilience, potentially ensuring the continued adaptability of the human lineage.

## Methods

### Evolutionary alteration of sequences in human immune-cell enhancers

**Creating the list of putative immune-cell enhancers.** Coordinates of putative enhancers for human CD34⁺, CD4⁺ T cells, CD8⁺ T cells, B cells, CD19⁺ cells, monocytes, macrophages, and dendritic cells were obtained from the EnhancerAtlas2.0 database[23] (http://www.enhanceratlas.org/), as well as for lymphoblastoid cell lines from Garcia-Perez et al.[24]. EnhancerAtlas 2.0 BED file coordinates were

converted from the human genome GRCh37 assembly to the human genome GRCh38 assembly using the UCSC LiftOver tool[29] with default parameters and the "hg19ToHg38.over.chain" file. Coordinates overlapping between cell types were merged, and all enhancers were pooled using BEDtools::mergeBed (v2.25.0). To further select regions with accessible chromatin, ATAC-sequencing (ATAC-seq) data for different immune cell types, including both progenitor and mature cells, stimulated or not, from Garcia-Perez et al.[24], Corces et al.[25], and Calderon et al.[26] were reanalyzed. For each separate dataset, ATAC-seq reads from the same cell types (separately for stimulated and unstimulated conditions) were merged using Samtools v1.3 and aligned to the GRCh38 genome using Bowtie2 v2.2.9 (parameters: -q -N 1 -p 8)[5]. Peaks were called using MACS2 v2.1.1 (parameters: -g hs -q 1e-5)[6], and only robust ones with a five-fold enrichment over background were selected to identify putative enhancers with accessible chromatin. Regions containing at least one ATAC-peak summit were termed immune-cell enhancers and combined for further analysis.

**Stratifying human immune-cell enhancers according to the evolutionary rate of sequence divergence.** To compare the DNA of human immune-cell enhancers with that of the Rhesus macaque and Chimpanzee, coordinates from the human GRCh38 genome were converted to the Macaca mulatta (RheMac10) and Pan troglodytes (PanTro6) genome assemblies using the LiftOver tool with a minimum match threshold of 0.97. Regions that were reciprocally mapped between human, chimpanzee, and macaque genomes via three-way LiftOver were classified as static (three-species orthologs). Regions that converted bidirectionally between human and chimpanzee but not between human and macaque were classified as intermediate. Regions that failed to map to both chimpanzee and macaque genomes were designated as rapid, reflecting human-specific sequence variation. Intermediate and rapid enhancers were collectively referred to as dynamic.

**Quantifying single-nucleotide substitution rate since the divergence from macaque.** To quantify the single-nucleotide mutation rate per enhancer within each group since the divergence from macaque, human DNA was mapped against macaque DNA using the BLASTn tool (Nucleotide-Nucleotide BLAST 2.13.0 +)[61] with a specific output format parameter -outfmt "6 qseqid sseqid pident length mismatch gapopen qstart qend sstart send evalue bitscore qseq sseq". Sequences of static enhancers were compared to their orthologs in the macaque. For dynamic enhancers, quasi-orthologs in macaque were identified by converting coordinates from the human GRCh38 assembly to the RheMac10 assembly with a -minMatch parameter of 0.5. The proportion of mismatched single nucleotides between human and macaque in regions alignable without gaps was identified for each enhancer. To control for potential misalignments around gaps, coordinates of mismatches and gaps were retrieved from the "subject" sequence in the BLASTn output column *sseq* (aligned part of subject sequence), and for each mismatch, its distance to the nearest gap boundary was computed.

### Identification of enhancers associated with the immune response

The activity-by-contact (ABC)[39] maps were used to predict enhancer interactions with genes associated with the immune response. These genes were identified by Hawash et al. [7] based on the ex vivo stimulation of human blood cells for four hours with LPS or GARP. The file "AllPredictions.AvgHiC.ABC0.015.minus150.ForABCPaperV3.txt" was downloaded from https://www.engreitzlab.org/resources/. Enhancer coordinates in immune cell types were lifted to the human genome GRCh38 assembly using the UCSC LiftOver tool with default parameters and the "hg19ToHg38.over.chain" file. If enhancers from the present study overlapped with those from the ABC source by at least 1 bp, they were considered to represent the same region.

## Analysis of transposable elements (TEs)

**Drawing trajectory of primate-specific TE (pTE) sequence acquisition over evolutionary time.** TE coordinates and annotations were retrieved from the human hg38 RepeatMasker (Smit, A.F.A., Hubley, R., & Green, P. RepeatMasker Open-4.0.5 Repeat Library 20140131 available at http://www.repeatmasker.org). To identify primate-specific TEs (pTEs), TE clade information was extracted from the file "20141105_hg38_TEage_-with-nonTE.txt" downloadable from the TEanalysis tool (https://github.com/4ureliek/Teanalysis). pTEs gained since the divergence from macaque were identified by mapping their coordinates to genomic gaps in chimpanzee (PanTro6) and macaque (RheMac10) relative to the human genome (GRCh38) using BEDtools::intersectBed tool (-f 0.9). These gaps (not alignable regions) were obtained using the UCSC maf-NoAlign tool along with the syntenic alignments from the MAF files hg38.panTro6.synNet (https://hgdownload.soe.ucsc.edu/goldenPath/hg38/vsPanTro6/) and hg38.rheMac10.synNet (https://hgdownload.soe.ucsc.edu/goldenPath/hg38/vsRheMac10/). For Fig. 1d, e, a pTE sequence overlapped a gap in macaque (RheMac10) or chimpanzee (PanTro6), regardless of the aligned length, and was classified as gained since the divergence from macaque or chimpanzee, respectively. Otherwise, it was classified as shared. A similar analysis was performed on enhancers normalized to ±500 bp and ±1000 bp from their midpoints.

**Analyzing TE subfamily enrichment in enhancers.** Parsing was performed at both the individual copy and subfamily levels using hg38 RepeatMasker output file with the custom parseRM.pl script[62]. TE subfamily abundance within the enhancer group was compared to that of the genome using the TE-analysis_pipeline.pl script (https://github.com/4ureliek/Teanalysis) with default parameters (10 bp overlap between TE and enhancer). Enrichment significance (by copy count and length coverage) was assessed using hypergeometric and binomial tests (adjusted $P < 0.01$). Two additional controls were applied to account for potential biases from enhancer size and the LiftOver procedure. Random regions not annotated in EnhancerAtlas2 and matching to immune-cell enhancers by number and length distribution were shuffled 1000 times using BEDtools::shuffleBed. These random regions were split into three sets, (i) mirroring the individual enhancer groups in both number and length distribution, and (ii) using the same LiftOver-based procedure previously applied to the immune-cell enhancers. Similar statistical tests were applied to the (i) and (ii) control regions to define TE subfamily enrichment. TE subfamilies were defined as enhancer-enriched if significantly overrepresented in enhancers versus the genome by both hypergeometric and binomial tests (based on counts and coverage) but not enriched by either test in size- and LiftOver-group-matched random regions (with a tolerance of ≤5%). In Fig. 1f, a pTE copy from the enriched subfamily was classified as gained if ≥90% of its length overlapped a genomic gap in the macaque or chimpanzee genomes.

**Quantifying pTE contribution to accessible chromatin.** ATAC-seq peaks from distinct immune-cell populations were classified as shared if ≥50% of their length aligned to the RheMac10 genome and as novel is if >50% of their length overlapped gaps in RheMac10 relative to GRCh38, as accessed by BEDtools::intersectBed (-f .5). ATAC-seq peaks were considered Alu-derived or pERV-derived if ≥50% of their sequence was composed of Alu or pERVs sequences, respectively. Among the novel ATAC-seq peaks, those in which ≥50% of the gap originated from Alu or pERV sequences were defined as novel Alu-derived or novel pERV-derived peaks, respectively. The proportion of Alu- or pERV-derived peaks, whether shared or novel, was quantified relative to the total number of shared or novel peaks, respectively. The spatial distribution of ATAC-seq signal relative to Alus and pERVs that overlapped ATAC-peak by ≥50% was calculated using deep-Tools_2.2.4::computeMatrix (scale-region) function and plotted using deepTools_2.2.4::plotHeatmap function.

## Transcription factor binding site (TFBS) analysis

**Quantifying TFBS enrichment in enhancers.** TFBS enrichment analysis across the three individual enhancer groups was performed using the Homer (v4.11) "findMotifsGenome.pl" tool with the parameter "-size given", utilizing a Homer custom TFBS database (http://homer.ucsd.edu/homer/motif/motifDatabase.html) and a significance threshold of $P < 0.01$. To compare the enrichment or depletion of inflammation-related TFBS in dynamic enhancer groups compared to static enhancers (set as a background), genomic coordinates of IRF and NF-κB-related TFBS were retrieved from the human JASPAR2022 USCS Transcription Factors Tracks (http://expdata.cmmt.ubc.ca/JASPAR/downloads/UCSC_tracks/2022/hg38/).

A two-tailed hypergeometric test was used to assess TFBS enrichment or depletion (adjusted $P < 1.00e{-}10$).

**Identifying TFBS gained since the divergence from macaque.** In Fig. 2, TFBS were defined as shared or gained based on the 100% overlap with genomic gaps in the macaque (RheMac10), gibbon (NomLeu3), or chimpanzee (PanTro6), respectively, compared to the human genome GRCh38. Gaps were identified as described above.

**Identifying pTE contribution to the TFBS creation since the divergence from macaque.** Enhancer-linked pTEs with fully embedded TFBS were identified by intersecting pTE coordinates with the TFBS coordinates from the JASPAR2022 USCS tracks using BEDtools::intersectBed with the -F 1 option. The proportion of each TFBS class derived from distinct pTE biotypes (Alu, pERV, pLINE, pDNA, and SVA) or non-TE sequences in enhancers was calculated separately for shared and great-ape-specific TFBS relative to their total number in enhancers.

**Analyzing TFBS within enhancer-associated ChIP-seq peaks.** ChIP-seq peak coordinates for IRF1 and NF-κB family proteins were collected from ENCODE and REMAP2022 databases (https://doi.org/10.1093/nar/gkab996). Only peaks overlapping immune-cell enhancers by ≥50% were retained. ChIP-seq peaks for NF-κB family proteins (NF-κB1, NF-κB2, RELA, and RELB) were combined using BEDtools::mergeBed with default parameters. Genomic coordinates of NF-κB motifs (NF-κB1_MA0105.4, NF-κB2_MA0778.1, RelA_MA0107.1, and RelB_MA1117.1) and IRF1_MA0050.2 motif from JASPAR2022 USCS tracks were intersected with the corresponding TF ChIP-seq peaks using BEDtools::intersectBed (-f 1), requiring the motif to be fully embedded (100% overlap) within a peak to be defined as bound TFBS. pTE-derived motifs were determined based on a 100% overlap between the motif and the pTE sequence. For Fig. 3b, the proportion of bound pTE-derived motifs versus unbound motifs in enhancers was quantified across pTEs grouped by biotype (Alu, pERV, pLINE, and pDNA). For Fig. 3c, the contribution of different enhancer sequences (pTE biotypes and non-pTEs) to TFBS binding was expressed as the fraction of bound motifs derived from a specific sequence type relative to all bound motifs for the same TF class within enhancers.

**Quantifying TFBS enrichment in enhancer-associated ChIP-Seq peaks.** Inflammation-related transcription TFBS enrichment analysis within enhancer-associated NF-κB peaks or IRF1 peaks was performed using the Homer (v4.11) "findMotifsGenome.pl" tool with the parameter "-size given," utilizing customized JASPAR2024 CORE vertebrates redundant position weight matrices (PFMs) and a significance threshold of $P < 0.01$.

**Enrichment of Alu- and pERV-derived TFBS motifs in ChIP-Seq peaks.** The enrichment or depletion of Alu-derived NF-κB-related motifs within NF-κB family proteins ChIP-seq peaks or IRF1 motif within IRF1 peaks, an arbitrary background was created using ChIP-seq peaks uniquely attributed to the unrelated transcription

factors ETS1 and RUNX1 from REMAP[63]. A two-sided Fisher's exact test ($P < 0.01$) was used to compare the ratio of Alu-derived or pERV-derived NF-κB-related motifs (NF-κB1_MA0105.4, NF-κB2_MA0778.1, RELA_MA0107.1, and RELB_MA1117.1 pooled together) to non-Alu-derived or non-pERV-derived counterparts within NF-κB ChIP-seq peaks versus their ratio within ETS1- or RUNX1-exclusive peaks. The same statistical analysis was applied for IRF1 motifs in IRF1 ChIP-seq peaks.

**Identifying TFBS in pTE consensus sequences.** Consensus sequences for Alu and pERV subfamilies most abundant in NF-κB or IRF1 ChIP-seq peaks were extracted from the Dfam collection of consensus models (https://dfam.org)[44]. "Dfam.embl" file was downloaded and converted to FASTA format using web-based "embl_to_fasta" tool. FIMO tool (5.1.1) was used to scan TE subfamily consensus sequences for IRF1 and NF-κB-related motifs, using PWMs from JASPAR2022 and thresholds $q < 0.0001$ (stringent match) and $q < 0.001$ (permissive match).

**In silico prediction of TFBS transcription factor binding affinity.** Nucleotide sequences of Alu- and pERV-derived NF-κB1 (MA0105.4) and IRF1(MA0050.2) in enhancers were retrieved using the Biostrings::getSeq function in R (version 4.2.2) and the BSgenome object "Hsapiens.UCSC.hg38" (R package version 1.4.5). These TFBS were converted to the chimpanzee (PanTro6) and macaque (RheMac10) genomes using the UCSC LiftOver tool with a -minMatch parameter of 0.5 to obtain motifs shared between human and other primates. TFBS sequences shared between chimpanzee and macaque were extracted using the Biostrings::getSeq function with the BSgenome.Ptroglodytes.UCSC.panTro6 and BSgenome.Mmulatta.UCSC.rheMac10 objects. TF binding affinity of TFBS was quantified using the TFBSTools::searchSeq function in R[48]. To assess the evolution of TFBS binding affinity since the human-macaque divergence, affinities of shared TFBS were compared between human and macaque or chimpanzee. For great-ape-specific TFBS, comparisons were made between human and chimpanzee sequences, as well as between human TFBS in enhancers and subfamily consensus sequences. The difference (Δ) in TFBS binding affinity was calculated by comparing maximum scores generated by TFBSTools::searchSeq in R. The trajectory of TFBS binding affinity was classified as increasing if $\Delta \geq 2$, decreasing if $\Delta \leq -2$, and neutral otherwise.

## Positive selection in modern human populations
**Positive selection at immune-cell human enhancers.** To estimate how immune enhancers have been targets of positive selection in recent human history, we focused on Central Europeans from Utah (CEU), Yoruba from Ibadan, Nigeria (YRI), and Southern Han Chinese from Hong Kong (CHS) populations from the 1000 Genomes Project, representing European, African, and East Asian ancestries, respectively. For each population, allele age and frequency data, as well as P-values indicating evidence of positive selection calculated using Relate, were downloaded from Zenodo. (https://zenodo.org/records/3234689).

We then measured evidence of positive selection at each variant by combining two orthogonal metrics:
(i)   The Relate P-value for positive selection[50] that contrasts the age of the derived allele (as inferred from local haplotypic patterns) to that of genome-wide SNPs matched for allele frequency, allowing to detect rapid increases in derived allele frequency.
(ii)  An empirical P-value derived from the population branch statistic (PBS), that captures population-specific changes in allele frequency[49]. For each population, PBS was computed based on Reynold's FST estimates[64], using the other two populations as control and outgroup. For each population, genome-wide PBS values were then ranked across all common variants (minor

allele frequency >5%), and each variant $v$ was assigned an empirical P-value $p_v$, defined as the percentage of common variants with a PBS value greater than $PBS(v)$.

For each variant and population, the two p-values were then combined using Fisher's method to obtain a combined P-value of positive selection. For each enhancer, we next applied Sidak's multiple testing correction across all variants overlapping the enhancer and all three populations and focused on the variant and population with the lowest adjusted p-value to obtain a single p-value of positive selection per enhancer. Finally, we applied Benjamini-Hochberg (BH) multiple testing correction across all enhancers and applied a 5% FDR threshold to define the set of enhancers evolving under positive selection.

Differences in the percentage of selected enhancers across groups of enhancers were tested using Fisher's exact test. For each group, we derived the 95% confidence interval of the percentage of enhancers under positive selection using the *binom.test* R function.

**SNP effect on TFBS binding affinity.** Positively selected SNPs (PS-SNP) within immune-cell enhancers were defined based on positive selection signals with $P < 0.01$. SNP counterparts evolving neutrally within enhancers were defined based on selection signals with $P > 0.5$. Binding affinity comparisons between derived and ancestral alleles were performed using TFBSTools::searchSeq R function as above.

**Genome coverage by different sequence types.** For Fig. 4c, RepeatMasker output was customized using TE clade information, classifying TEs into primate-specific groups according to biotype (Alus, pERVs, pLINE, pDNA, SVA) and ancient TEs (pooled). This file was processed using the parseRM.pl script (https://github.com/4ureliek/Teanalysis) to generate coverage of each primate group and ancient TEs in genome. The output file, <RMout.out > .parseRM.all-repeats.tab, reported the genomic coverage for each group. Overlapping regions between different sequence types accounted for 0.02% of the total and were excluded from the analysis. Coverage in enhancers was further calculated using TE-analysis_pipeline.pl script (-TEov 1) (https://github.com/4ureliek/Teanalysis), with input from the parseRM.pl outputs and the same customized file. Coverage of enhancers by non-TE sequences was defined as 100%-SUM(TE-coverage).

## Identification and characterization of inflammatory disease enhancers (IDEs)
Genes associated with the main inflammatory and autoimmune disorders were downloaded from the DisGenNet database (https://www.disgenet.com), and those overlapping with at least 30% of the conditions were selected. Their putative enhancers were defined using ABC maps (see above) for immune cell populations. Among these, inflammatory disease enhancers (IDEs) were identified as regions that overlap (by ≥ 50%) ChIP-seq peaks for any of the following transcription factors: IRF1, NF-κB1, NF-κB2, RELA, or RELB, as determined using BEDtools. TE subfamilies enriched in IDE compared to other enhancers (set as a background) were identified using the same statistical approach as for TE subfamilies enriched in enhancers (see above). The proportion of IDEs associated with genes involved in great ape- or human-specific transcriptional immune responses (from Hawash et al.[7]) was quantified based on their ABC-defined links.

## Data visualization
Data was visualized in the genome browser IGV 2.8.13.

Global selection patterns were visualized using the Geography of Genetic Variants Browser.

PheWas plots were created using data from the Open Targets Genetics site, choosing association of traits from FinnGen, UK Biobank, and GWAS Catalog (https://www.genetics.opentargets.org).

## Reporting summary

Further information on research design is available in the Nature Portfolio Reporting Summary linked to this article.

## Data availability

All datasets used in this study are publicly available from the repositories cited in Supplementary Data 1 and 6. All data generated in this study are provided in the Supplementary Data and Source Data files. Source data are provided with this paper.

## Code availability

Analyses were performed using open-source software: BEDtools v2.25.0, MACS2 v2.1.1, Samtools v1.3, Bowtie2 v2.2.9, UCSC toolkit, hg38 RepeatMasker with the parseRM.pl script from TEanalysis, deepTools v2.2.4, HOMER v4.11, FIMO (MEME suite) v5.1.1, TEanalysis (https://github.com/4ureliek/Teanalysis). *P*-values indicating evidence of positive selection calculated using Relate, were downloaded from Zenodo. (https://zenodo.org/records/3234689). R version 4.2.2 (2022-10-31): platform: x86_64-conda-linux-gnu (64-bit) running under Ubuntu 16.04.7 LTS. Attached packages: BiocManager_1.30.23, GenomicRanges_1.50.2, GenomeInfoDb_1.34.9, IRanges_2.32.0, BiocGenerics_0.44.0, Biostrings_2.66.0, S4Vectors_0.36.2, TFBSTools_1.36.0, BSgenome_1.66.3, BSgenome.Hsapiens.UCSC.hg38_1.4.5, BSgenome.Ptroglodytes.UCSC.panTro6_1.4.2, BSgenome.Mmulatta.UCSC.rheMac10_1.4.2, dplyr_1.1.4, ggplot2_3.5.1, and ComplexHeatmap_2.14.0. Additional information is available upon request.

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

## Acknowledgements
The authors thank Lluis Quintana-Murci, Camille Berthelot and all members of the Amigorena team for insightful discussions. Primate icons are created in BioRender. Zueva, E. (2025) https://BioRender.com/8d7b1bb (agreement number: KV28PQR9HF).

## Author contributions
E.Z. conceived and designed the project. E.Z., M.Y. and M.R. developed the methodology and analysed the results. E.Z., S.A. and M.R. acquired funding. E.Z. wrote the original draft. E.Z., M.R., S.A. and M.Y. reviewed and edited the manuscript. All authors read and approved the final manuscript. This work is supported by the Agence Nationale de la Recherche; ANR-23-CE14-0017-02 (E.Z.), ANR-10-IDEX-0001-02 PSL (S.A.), and ANR-22-CE12-0030-01 (M.R.), by Mnemo Therapeutics and the Center of Clinical Investigation, CIC IGR-Curie 1428 of Inserm (M.Y.), and Program France 2030 launched by the French Government through the LabEx DCBIOL (S.A.).

## Competing interests
The authors declare no competing interests.
