## [Peer Review file · Nature Communications]

Transposon invasion of primate genomes shaped human inflammatory enhancers and susceptibility to inflammatory diseases

Corresponding Author: Dr Elina Zueva

Version 0:

Reviewer comments:

Reviewer #1

(Remarks to the Author)

Interesting study that uses comparative genomics and population genetics to explore the role of primate-specific TE to the human gene regulatory network associated with immunity. Although similar small-scale observations have been done before, the objective of a comprehensive study on this makes a lot of sense.

That being said, I do have several technical concerns with the study as-is.

Comments:

3Kb seems very large for enhancers (Ext Fig 1c). Is it possible you are merging different enhancers together? What is the density of ATAC-seq peaks within these regions? Are they uniformly distributed, or mostly localized in the center? What would happen to your analysis if you were more stringent in the merging step? Or if you selected only the center of the enhancers? See also next point.

I'm worried about the liftOver step which will also be affected by the length of the enhancer intervals. Are short versus long enhancers more or less likely to be successfully liftOver? If you were to length-correct all enhancers to be of the same size (e.g. 1Kb), would it change your results? When you say "While dynamic regions were not alignable to the macaque genome at a given threshold, lowering it to 50% alignability identified quasi-orthologs for 96% of them, indicating that the alteration of ancestral sequences is more prevalent than the emergence of enhancers from entirely novel DNA", I once again worried that all of this is affected by the fact that you have large regions at that your ortholog mapping is not really specific to the actual enhancers (ie location of the ATAC-seq peaks themselves).

I'm also a bit worried that this analysis based on enhancers should mimic what would be seen with the ATAC-seq peaks themselves. Is that the case? Would you see the same thing if you were to use some of the original datasets used to create your "large" enhancer regions? I'm just worried that you're bringing genomic noise with your enhancers and that because you then classify things based on lack of ortholog mapping with liftOver, you might be enriching for young Alus in the dynamic enhancers. Is that what you did in Ext Fig 1a-b? Those numbers look much more comparable between ERVs and Alus and are much lower than in your general enhancers. Again, I'm just worried that your analysis is not very specific because your enhancers are too large.

In Fig 1c, could the increase rate of mutations come from misalignment around gap regions?

For the analyses looking at overlap with pTEs (Fig 1d-e), I'm also worried that enhancer size is a factor. Can you repeat the analysis by adjusting the size of the enhancers? In Fig 1e we see that pTEs are really only contributing a minor components of the enhancers, to me this is a concern as to whether or not the peaks detected have anything to do with pTEs.

I think Fig 1f is partially circular. The random background is ok but you also need to control for the liftOver step. What happens if you liftOver your random set and classify them as you did for the observed data? That's the actual control you need to use to measure enrichment per enhancer group. This also applies for Fig 1g. If I understand correctly, these enriched families are also the basis for Ext Fig 2 which is also a problem because it doesn't control for the liftOver step. Ext

Fig 2b versus 2a shows that the strong Alu enrichment is weaker than the one in ERVs in terms of %. The big jump in the t-test comes from the fact that the background (sequences missing in macaques) are themselves very much enriched for Alus. In sum, it would be good to also show the level of enrichment but controlling for the liftOver step.

Moving to Fig 2, once you identify motifs that are enriched, could you show that these motifs are found in the ATAC-seq peaks, not just your broad putative enhancer regions? Can you show that the overlap is significant?

For Fig 3, it's great that you look at TFBS actually in the pTEs but how many of those are in actual peaks? Once again, I'm worried that your enhancer are too big and you're basically pulling in pTEs (which have non-functional TFBS) for the ride. Your Fig 3b does that better and we see once again that it's the ERVs and not the Alus that show the most interesting pattern (my guess is that all of this once again has to do with the liftOver step of large regions). It would be good to compare the aggregate location of peaks and motifs relative to pTEs. I suspect that the ERVs also show a more "centered" distribution compared to Alus, many of which might just be "near" the peaks. For Fig 3e, the consensus-based approach to determine the presence of the motif doesn't always work, a better approach would be to use a method as in Carter et al. LTR7 study. For Fig 3f, was this analysis restricted to TFBS with peaks?

For Fig 4, again it would be good to have a better control here. Could you perform the same analysis for other regions with Alus that were not found in the immune enhancers? Going back to the liftOver control I suggested above, it would be good to see if there's a difference but that control and these enhancer regions.

Minor comments:

In Ext Fig 1a, could you add the number of regions in each column/row?

Reviewer #2

(Remarks to the Author)

The authors investigated the evolutionary influence of transposable elements, and specifically Alu elements and endogenous retroviruses, on human immune-cell enhancers and inflammatory responses. Using comparative genomics, they traced enhancer sequence changes back to macaques and identified the role of transposons in reshaping NF- κ B and IRF1 binding motifs, particularly in great apes. They found that Alu-derived motifs often show increased binding affinity after the human-macaque split and are enriched for positive selection in humans, especially in enhancers associated with chronic inflammatory diseases. Their study highlights how transposon invasions uniquely shaped primate immune adaptation and continue to influence human inflammatory potential. The paper presents interesting and potentially compelling findings. However, to be accepted for publications, it requires several clarifications on how some of the analyses were performed, and on the rationale behind some of the analysis.

Specifically:

- Can the authors elaborate more about the immune-cell enhancers? Of the 60k enhancers, how many of them were present in all the cell types investigated? Or in how many cell types? Was there a threshold of number of immune cell types in which the enhancer was supposed to be detected? Or was one cell type enough?

- Identification of pTE-derived enhancers: can the authors explain more in detail in the results section how much overlap between enhancers and TEs was required to affirm that an enhancer was TE derived? Is this based on the TE_ANALYSIS tool (so is this significant enrichment) or simple overlap? And in the latter case what fraction of the enhancer was required to overlap a TE? A few sentences below they mention that the pTEs contribute to 2-3% of the length of the enhancer. But given that TEs represent half of the human genome, wouldn't this be significantly less than expected by chance? In the methods they mention a 90% overlap required for the gap analysis (and 50% for the ATAC-seq peaks), but what about all other comparisons? I am not implying that the analysis was not performed correctly, but simply that the result section would benefit from much more detailed description of the approach.

- Again, in the sentence: Over 70% of dynamic regions overlapped with pTEs from these enriched subfamilies, compared to less than 50% of static regions, what fraction of the length of the enhancer was set as threshold for being defined as "overlapping a TE"?

- On page 7, the authors write that Alu elements are the most abundant transposons in the human genome, but that is incorrect. LINE elements are, by far, the most abundant.

- Why did the authors focus on Alu and ERVs, not considering LINEs and SVAs? SVAs in particular are considered to be a major source of human-specific enhancers. Could they provide more rationale on the choice of investigate TEs?

- Page 9: the sentence "with gibbon added as a lesser ape to identify great-ape-specific gains". Could the authors clarify what this means?

- When the authors state that "pTEs made significant contributions to "shared" TFBS, accounting for 10% to 50% of the sites": how does this compare to random expectation? Is it more than expected by chance? Was there a p-value calculated?

- Regarding the CHIP-seq publicly available datasets (ENCODE) that the authors used: what was the read length? Were the reads paired ends? Were only uniquely mapped reads retained? These are key parameters to be implemented to be able to properly map reads on TE regions .
- Based on the methods, it seems that the authors did not correct for multiple testing (i.e no FDR) in their enrichment analysis performed with the TE analysis pipeline tool. Could the authors elaborate on why?
- In the methods pLINA should be pLINE?

Reviewer #3

(Remarks to the Author)

This study from Elina Zueva and colleagues finds an important and innovative evolutionary details of primate endogenous retroviruses (pERVs) and Alu elements. pERVs are remnants of ancient viruses that infiltrated the genomes of our primate ancestors and are now 8% of human genome. Initially, these viral invaders behaved like parasites, but over time they seeded our DNA with special sites that could attract immune regulators such as the IRF1 protein. In contrast, Alu elements, tiny, highly abundant DNA sequences (1 million copies) that make up to 10 % of our genome have evolved to favor the binding of NF- κ B, a protein that is crucial in managing inflammation. Although a single Alu element might have a small effect, the cumulative influence of thousands of them can dramatically impact how our genes respond to stress or infection. One of the novel findings of this research is the discovery that these elements are not static relics; they have been continuously refined by evolution. In particular, the study shows that specific segments within Alu elements have evolved precise NF- κ B binding motifs. For example, a distinct Alu-derived NF- κ B1 motif (referred to as MA0105.4) underwent rapid expansion in the common ancestor shared by humans and chimpanzees. This rapid expansion suggests that these elements provided ready-made genetic tools during periods of rapid environmental change, allowing our ancestors to fine-tune their inflammatory responses swiftly and effectively. This adaptive mechanism, forged over millions of years, not only explains the rapid evolution of our inflammatory responses but may also account for the genetic basis of susceptibility to inflammatory diseases in modern humans. Ultimately, these ancient DNA elements continue to contribute to our resilience by providing a vast, dynamic reservoir of genetic variability that can be drawn upon to meet new challenges. I like this study as it gets close to answer why there is a remarkable proliferation of Alu elements in the primate genome. It suggests that these elements may have undergone positive selection driven by their potent enhancer activity during immune cell stimulation, thereby contributing significantly to the host's regulatory responses upon pathogen invasion. This work indeed represents a significant advance over previously published studies. Thus, this line of advancement has enormous potential to outreach plenty of genome biology researchers, chromatin and transposon audiences. Overall, although preliminary, the results are interesting and worthy of publishing. At this point, however, several issues arise that need further clarification and analysis before I consider this study complete and make me less optimistic about this work getting published in its current form.

Major Concerns:

1. My major concern is that the manuscript does not adequately address its limitations. For example, the authors use activity-by-contact (ABC) maps to infer looping interactions between TE-derived enhancers and gene promoters. However, these maps only show physical proximity and do not establish a functional regulatory relationship. Moreover, it is unclear which specific genes are implicated and how these genes contribute to the innate immune response. A more thorough discussion identifying the gene sets and either referencing existing functional data or proposing targeted experiments would greatly strengthen the manuscript's premise. With functional data, I mean genetic experiments where the study has rescued the phenotype e.g., MER41B from PMID: 26941318.
2. Page 7, last sentence: The authors suggest that Alu insertions drive the formation of open chromatin. However, the evidence supporting this claim remains ambiguous. It is equally plausible that Alu elements preferentially integrate into preexisting open chromatin regions rather than creating novel enhancers. In fact, Extended Data Figure 2C appears to support this alternative interpretation, showing that ATAC-seq signals are similarly distributed in the neighboring regions of Alu insertions. A more detailed discussion of these observations to distinguish between these scenarios, would significantly strengthen the authors' conclusions.
3. On Page 10, Authors declare the Modus operandi that these enhancers regulating conserved and inflammation aspects of immune response - a claim that currently rests solely on computational analyses. Such a substantial conclusion warrants further functional validation. Specifically, experiments such as ChIP-STARR-seq would be necessary to confirm the enhancer activity of these Alu elements, followed by genetic perturbation studies to demonstrate that disruption of these enhancers leads to dysregulation of target gene expression and impacts the associated phenotype. I do not recommend that the authors should provide additional experimental evidence but they should tone down the claim.
4. Page 11 states that TE enhancers are entirely embedded within the corresponding ChIP-seq peaks which is the basis for selecting these enhancers. However, it remains unclear whether these enhancers overlap specifically with the peak summits and whether the sequences flanking these summits are enriched for the expected transcription factor binding motifs. I recommend that the authors provide a detailed analysis of the spatial distribution of TE enhancers relative to the ChIP-seq peak summits and assess motif enrichment in these critical regions. This additional information would substantially bolster the evidence for their proposed functional role.

5. Page 18: Authors write and claim “enhancers enriched in Alus suggest their key role in the evolutionary shaping of the human inflammatory response”. This is a provocative claim based on evidence of correlation from neighbor genes and gene ontology is substandard in the field. The investigations presented here are exploratory. Authors should refrain from making bold conclusions based on association analysis of TEs and genes and/or biological pathways unless the association is already published with a robust, reliable and reproducible set of experiments. I would encourage authors to re-write these sections with keeping the limitations of the gene ontology (DisGeNet in this case) and correlational tools in their conclusions.

6. The reanalysis of data on Page 31: The manuscript relies on ATAC-seq data from Corces et al. (Nat. Gen. 2016), which derives from selected and some cancer cell types, yet it remains unclear why the dataset from Calderon et al. (Nat. Gen. 2018) which encompasses ATAC-seq and RNA-seq profiles in the atlas form for diverse human T and B cell types in both resting and activated states was not incorporated. Given the physiological relevance of the Calderon dataset for assessing chromatin accessibility in immune cells, its inclusion could potentially provide a more robust validation of the observed enhancer dynamics. I recommend that the authors clarify their rationale for excluding this dataset or consider integrating these data to further substantiate their conclusions.

7. Page 32: Regarding target gene selection, it is crucial to distinguish whether a given enhancer functions as the sole regulatory element for its target gene or operates alongside multiple enhancers. Genes often possess multiple enhancers, including primary and shadow enhancers, which can act redundantly to ensure robust gene expression. Therefore, attributing gene regulation to a single enhancer without considering the broader enhancer landscape may lead to oversimplified conclusions. I recommend that the authors assess the enhancer redundancy within the genomic context of each target gene to accurately interpret the regulatory mechanisms involved.

8. Methodology on Page 33: The authors analyze the enrichment of TE subfamilies within enhancer regions by normalizing the length of these TEs but do not address how TE length may influence this enrichment. Longer TEs, such as full-length proviruses (~7 kb), might be underrepresented in enrichment analyses compared to shorter elements like solo long terminal repeats (LTRs) or Alu elements, which are only a few hundred base pairs long. This size discrepancy could lead to a bias favoring the detection of shorter TEs as enhancers. While I do acknowledge that it is essential to consider the impact of TE length on enrichment analyses, the longer TEs e.g., LTR12, HERVH, etc., that have a strong cis-regulatory potential especially have antiviral activities might be underrepresented in the analysis.

Minor Comments:

1. Authors should discuss why they did not observe THE1B elements with NFkB binding and cite the relevant literatures. THE1B was shown to have high chromatin accessibility in T-cells with NFkB motif and binding (PMID: 30381291, PMID: 39988678). Of note, PMID: 39988678 is my paper and I do not encourage authors to cite if they do not reason its relevance.

Figure 1B: Are the differences statistically significant ?

Figure 1 C: Why do authors not perform a comparative analysis of enhancer regions with heterochromatin regions in pairwise manner which might help them calculating a relative frequency.

Figure 1F: Labels on Y-axis (the name of TE families) looks cluttered. You can decrease the font to make them more visible.

Page 7, line 14: Please do not pitch the lines as less abundant TEs are more regulatory from a review journal. It might be more likely that host would conserve few copies with strong regulatory activities but there are remarkable exceptions including the one being shown in this paper.

Page 7, line 22: The word co-option is not appropriate here. To show co-option, you would require a genetic and rescue experiments as I have mentioned above.

Version 1:

Reviewer comments:

Reviewer #1

(Remarks to the Author)

The detailed response from the authors has helped address all my concerns. A lot of new controls have now been added and confirm the robustness of the results.

In particular, I do appreciate the additional controls that were included to show the impact of the size of the enhancers and of the leftover step. A small point but my interpretation of the new Fig S1f is that changing the size of the enhancers does have a pretty significant effect, since it changes the classification of 25%-33% of the “inter” and “rapid” category. My suggestion would be to present this result in a more balanced way. I don't quite agree with “... yielded comparable classification patterns, indicating that the merging procedure did not introduced any significant bias”.

Reviewer #2

(Remarks to the Author)

The authors have addressed all my concerns, the manuscript can be accepted for publication.

Reviewer #3

(Remarks to the Author)

The authors have adequately addressed my comments, and the manuscript is much improved.

RESPONSE TO REVIEWERS

Reviewer #1 (Remarks to the Author):

Interesting study that uses comparative genomics and population genetics to explore the role of primate-specific TE to the human gene regulatory network associated with immunity. Although similar small-scale observations have been done before, the objective of a comprehensive study on this makes a lot of sense.

We thank the reviewer for the positive appreciation of our work.

That being said, I do have several technical concerns with the study as-is.

Comments:

3Kb seems very large for enhancers (Ext Fig 1c).

To our knowledge, enhancer-distinguishing histone mark deposition can span regions ranging from ~0.1 kb to over 100 kb, with a typical median of 2–4 kb. By saying, “*3kb is large for enhancers*”, the reviewer likely refers to the length of a minimal enhancer, defined as the shortest DNA region capable of driving gene expression. However, to our knowledge, enhancer-distinguishing histone mark deposition can span regions ranging from ~0.1 kb to over 100 kb, with a typical median of 2-4 kb. How many “minimal enhancers” can be contained within these larger *cis*-regulatory regions has not been formally and systemaically assessed.

Complex binding patterns of transcription factors, including those that bind nucleosomal DNA, and the presence of epigenetic regulators within regions marked by enhancer-distinguishing histone modifications highlight the importance of chromatin context, even though potent minimal enhancers can be detected through depletion assays. In particular, nucleosome-occluded enhancer DNA is crucial for epigenetic regulation of the enhancer itself.

Additionally, loci of cellular identity genes frequently contain large regions bound by transcriptional co-activators, broadly marked by H3K27ac, and enriched for in GWAS catalog SNPs, including those related to autoimmunity (PMID: 24119843, PMID: 24127591). It is unclear how to trim these larger regions without excluding biologically-relevant parts of *cis*-regulatory elements. Accordingly, our initial conservative approach was to analyse the full enhancer length, without trimming them to an uniform size. We have now extended our analyses to include additional controls, allowing to assess the impact of enhancer length variation on our main conclusions (Extended Data Fig. 1 and 2 and see below).

Is it possible you are merging different enhancers together?

We appreciate the opportunity to clarify this point. Indeed, in our effort to be concise, we weren't perfectly clear about the reason and the procedure for merging individual cell enhancers. Our approach to consider the union of all overlapping enhancers across cell types was driven by the observation that in 96% of cases, the final merged regions matched the largest overlapping individual coordinates, indicating that we captured the broadest boundaries of shared "regulatory hubs", which are frequently used by immune cells either in part or as a whole (revised Extended Data Fig 1e and below).

Moreover, the final size-distribution resembled that of individual cell types, with no shift toward longer elements but a reduced representation of shorter ones, consistent with the consolidation of overlapping coordinates.

Thus, merging did not generate artificially large regions but instead extended the boundaries of smaller ones to align with overlapping, broader regions.

To address any further concern regarding the merging of distinct enhancers, we have now added these additional quantifications (in revised Extended Data Fig.1 and 2) and clarified this point explicitly in the Results section.

What is the density of ATAC-seq peaks within these regions? Are they uniformly distributed, or mostly localized in the center?

To answer this question, we visualized the distribution of the ATAC signal (Extended Data Fig 1 and below), aligned with our final merged regions and their ± 1.5 kb flanking boundaries. As illustrated below for representative cell populations, the final enhancer regions are broadly accessible across different cell types, with higher signals in activated versus resting states. Importantly, the ATAC-seq signal delineates their boundaries, showing minimal adjacent noise, while the accessibility is sustained across regions' full length, despite signal peaking near the center. We, therefore, believe that the background introduced by the merging procedure is likely minimal and should not significantly affect subsequent enrichment analyses.

Notably, some regions initially not annotated as enhancers in specific cell types were nonetheless accessible in the corresponding cell types, suggesting a higher degree of regulatory promiscuity than we previously assumed, given that only 60% of our final regions were shared by at least two cell types (revised Extended Data Fig.1d).

That being said we agree with the reviewer that additional controls are essential when testing for TE or TFBS enrichment to account for potential background. We have incorporated these controls in the revised version (see below).

What would happen to your analysis if you were more stringent in the merging step? Or if you selected only the center of the enhancers? See also next point.

I'm worried about the liftOver step which will also be affected by the length of the enhancer intervals. Are short versus long enhancers more or less likely to be successfully liftOver?

If you were to length-correct all enhancers to be of the same size (e.g. 1Kb), would it change your results? When you say “While dynamic regions were not alignable to the macaque genome at a given threshold, lowering it to 50% alignability identified quasi-orthologs for 96% of them, indicating that the alteration of ancestral sequences is more prevalent than the emergence of enhancers from entirely novel DNA”, I once again worried that all of this is affected by the fact that you have large regions at that your ortholog mapping is not really specific to the actual enhancers (ie location of the ATAC-seq peaks themselves).

As discussed above, our definition of enhancers does not solely involve ATAC-seq peaks. We consider full-length regions marked by enhancer-distinguishing histone marks, as defined in the Enhancer Atlas.

To normalise for size while retaining maximum of core features, we corrected enhancer length to ± 1000 bp from midpoints. We assessed whether enhancer classification is maintained after such size-normalisation. Applying a similar liftOver, we observed that the majority of enhancers preserve their originally assigned category (revised Extended Data 1f and below). Shifts into other categories are mainly due to longer regions (including super-enhancers, which may contain areas with different degrees of conservation. Moreover, liftOver of individual cell type enhancers globally preserves the original categories (Extended Data 1f and below). For more controls, see further below.

f Re-classification of size-normalised enhancers based on LiftOver

Re-classification of individual cell type enhancers based on LiftOver

I'm also a bit worried that this analysis based on enhancers should mimic what would be seen with the ATAC-seq peaks themselves. Is that the case? Would you see the same thing if you were to use some of the original datasets used to create your “large” enhancer regions? I'm just worried that you're bringing genomic noise with your enhancers and that because you then classify things based on lack of ortholog mapping with liftOver, you might be enriching for young Alus in the dynamic enhancers. Is that what you did in Ext Fig 1a-b? Those numbers look much more

comparable between ERVs and Alus and are much lower than in your general enhancers. Again, I'm just worried that your analysis is not very specific because your enhancers are too large.

We believe that ATAC-seq peaks display distinctive features, reflecting their specific functions within broader regulatory modules that extend beyond regions of naked DNA. Accordingly, we do not expect enrichment patterns within these peaks to simply recapitulate those of the surrounding chromatin. Our previous study on mouse enhancers supports this interpretation (PMID: 32193341).

While our primary focus was on TE subfamilies enriched within full-length enhancers rather than individual ATAC-seq peaks, we agree that assessing enrichment in ATAC-seq peaks adds a finer-grained perspective. To address this, we examined TE enrichment at summits of ATAC-seq peaks from distinct immune populations (with TE subfamily counted only once if enriched in many).

As expected, ERV subfamilies were predominantly enriched at ATAC-seq summits, with a greater number of primate-specific ERVs enriched in peaks located within dynamic enhancers (revised Extended Data Fig. 3b and below). In contrast, Alu elements, although a subset exhibiting the strongest ATAC-seq signals (Extended Data Fig. 3c and below), did not reach statistical significance in this specific analysis.

However, when comparing the summits of dynamic enhancers to those of static enhancers, many Alu subfamilies emerge as enriched (revised Extended Data Fig.3b and below). We view this as further evidence that dynamic regions evolved differently from static ones, as here, only ATAC peak summits have been interrogated, and the analysis does not depend on size.

In Fig 1c, could the increase rate of mutations come from misalignment around gap regions?

For all enhancer groups, most mismatches are not adjacent to alignment gaps. Their density increases beyond ~50–100 bp from the nearest gap boundary, peaking between 100 and 300 bp, ruling out gap-related misalignment artifacts. (Extended Data Fig. 2a and below).

a Distribution of nucleotide mismatches relative to genomic gaps in macaque vs human

For the analyses looking at overlap with pTEs (Fig 1d-e), I'm also worried that enhancer size is a factor. Can you repeat the analysis by adjusting the size of the enhancers?

As discussed above, we are cautious about artificially adjusting enhancer sizes due to the potential biological relevance of their natural boundaries and the inherent challenges in defining a meaningful “adjustment.”

However, the point of the reviewer is pertinent as this type of analysis may help to understand whether flanks of enhancers are significantly different compared to central parts. We size-normalised enhancers to ± 500 bp and ± 1000 bp and performed the same quantifications, as in Fig. 1d and e. A similar trend was observed suggesting the pTEs were accumulated across the whole enhancer region (Extended Data Fig. 2b and c, and below).

In Fig 1e we see that pTEs are really only contributing a minor components of the enhancers, to me this is a concern as to whether or not the peaks detected have anything to do with pTEs. I think Fig 1f is partially circular. The random background is ok but you also need to control for the liftOver step. What happens if you liftOver your random set and classify them as you did for the observed data? That's the actual control you need to use to measure enrichment per enhancer group. This also applies for Fig 1g. If I understand correctly, these enriched families are also the basis for Ext Fig 2 which is also a problem because it doesn't control for the liftOver step.

Fig. 1e (left panel) shows that pTE sequence enhancer coverage ranges from 10% to 100%, with a median exceeding 20% for intermediate enhancers. We believe the impression that pTEs are “minor component” may stem from our insufficient emphasis on this coverage in the original figure. This has now been clarified and visually improved.

That said, we agree with the reviewer that an additional control for the liftOver step is warranted. Following this suggestion, we performed 1,000 iterations of random shuffling of non-enhancer regions matched in number and size distribution to our enhancer pool, and subjected them to the same liftOver strategy as in our main analysis. This followed by a similar three-group classification. We then performed TE subfamily enrichment tests in these three control groups using the same criteria as for enhancer regions.

We retained as “enhancer-enriched” the majority of subfamilies enriched in the initial analysis except AluY species, which were also enriched in these control regions. Importantly, AluS and AluJ subfamilies, most abundant in our enhancers, were not enriched in these control regions (5% tolerance, see the revised Supplementary Dataset 3) and, therefore, retained as specifically enriched in enhancers.

The list of enriched TE subfamilies has now been corrected by excluding those enriched in liftOver control regions (see revised Supplementary Dataset 3). Our revised Fig.1f (also see below) now includes the top TE subfamilies that are: i) enriched in each enhancer group according to the initial analysis, and ii) not enriched in this new controls for liftOver procedure.

As shown in the revised Fig.1g (and below), the association of enhancer groups with primate and ancient TEs remains broadly consistent with the initial analysis.

g
% of enhancers overlapping with enriched TE subfamily instances

Moreover, we performed a similar TE subfamily enrichment using enhancer size-normalization (± 500 bp and $\pm 1,000$ bp from the enhancer midpoint (Extended Data Fig. 2d and below). Similar to untrimmed regions, static enhancers remained enriched for ancient TE subfamilies, while dynamic ones favored primate-specific TEs, including Alus, confirming that this pattern persists across the entire enhancer span. Although enrichment levels of certain ERVs varied, as this analysis is not identical, the most abundant Alu subfamilies consistently remained enriched. The raw results of this analysis are now added to the Supplementary Dataset 3.

TE subfamilies enriched in size-normalised enhancers vs genome and not enriched in the matching random control regions

Ext Fig 2b versus 2a shows that the strong Alu enrichment is weaker than the one in ERVs in terms of %. The big jump in the t-test comes from the fact that the background (sequences missing in macaques) are themselves very much enriched for Alus. In sum, it would be good to also show the level of enrichment but controlling for the liftOver step.

The original Extended Data Fig. 2 was designed to illustrate that great-ape-specific Alu insertions are more likely to harbor accessible chromatin than older insertions and that this trend is more pronounced for Alus than for ERVs.

To further address the reviewer’s concern regarding the potential random association between Alu insertions and accessible regions located on novel DNA (absent from the macaque genome), we added a density plot of ATAC-seq signal across Alu sequences. It supports a non-random distribution, with higher read density overlapping with pTE elements (revised Extended Data Fig. 3c and below). This reduces the likelihood of their random overlap with peaks within evolutionary novel DNA (if we correctly interpreted the reviewer’s concern). Moreover, as mentioned above, specific Alu subfamilies remain enriched in our regions after applying additional control for LiftOver, showing that their association with enhancers is not random.

Moving to Fig 2, once you identify motifs that are enriched, could you show that these motifs are found in the ATAC-seq peaks, not just your broad putative enhancer regions? Can you show that the overlap is significant?

This is a very useful suggestion, which we overlooked by concentrating on ChIP-Seq peaks. We now analysed the proportion of enhancer TFBS fully embedded within ATAC-seq peaks, and also performed a classical TFBS enrichment test using the same Homer algorithm as for enhancers. We found that a significant proportion of TFBS in enhancers are embedded within accessible chromatin (ranging from 24% to over 50%; Fig.2b and below), with these motifs enriched within ATAC-seq peaks (revised Supplementary Dataset 4). We also revised Fig.2 to accommodate this information more effectively.

It is noteworthy that some transcription factors, such as for example, NF- κ B, bind both naked and nucleosome-bound DNA (PMID: 24086160). For these TFs, we may underestimate the number of functional motifs by analysing ATAC-seq only.

Of notice, we believe that the mere presence of TFBS within an overall favorable enhancer environment may reflect the enhancer's evolvable potential. For example, Alu elements have been proposed to represent proto-enhancers that gradually acquire active enhancer features, including TF binding potential, as suggested by *Su et al.* (PMID: 24703844).

For Fig 3, it's great that you look at TFBS actually in the pTEs but how many of those are in actual peaks? Once again, I'm worried that your enhancer are too big and you're basically pulling in pTEs (which have non-functional TFBS) for the ride.

Quantifications in Fig.3 (except for 3a) specifically refer to NF- κ B (and IRF1, now moved to Extended Data Fig. 6) motifs that are fully embedded within ChIP-seq peaks (100% overlap). Thus, these motifs are in *actual* peaks.

We believe that a comprehensive analysis of all ChIP-Seq for TFBS classes shown in Fig.3a, including those unrelated to inflammation, extends beyond the scope of this study and may detract from the focus on core inflammatory regulators. We hope that the additional analysis presented above, examining all TFBS enriched within ATAC-seq peaks, sufficiently illustrates their widespread accessibility, as reflected in the revised Fig.3a.

Your Fig 3b does that better and we see once again that it's the ERVs and not the Alus that show the most interesting pattern (my guess is that all of this once again has to do with the liftOver step

of large regions). It would be good to compare the aggregate location of peaks and motifs relative to pTEs. I suspect that the ERVs also show a more “centered” distribution compared to Alus, many of which might just be “near” the peaks.

We want to clarify that our intention is not to present ERVs and Alus in competition with one another. ERVs exhibit a robust pattern of transcription factor binding, as acknowledged in the main text, particularly in the revised version where we have added previously omitted THE1B elements, which were also abundant in NFKB ChIP-Seq peaks.

We agree with the reviewer that it is important to visualize aggregate reads on Alus and ERVs. We have centered NFKB ChIP-seq reads from different lymphoblastoid cells lines on Alu and ERV-derived TFBS rather than elements themselves because Fig.3 primarily focuses on TFBS quantification.

We observed that in most cases, both Alu- and pERV-derived NF- κ B motifs well-align with the highest ChIP-Seq signal intensity (revised Fig.3e and below). At the same time, Alu-derived NF- κ B motifs may show somewhat less sharp distribution compared to pERV-derived. A cohort of Alu-derived TFBS may play a distinct role in NFKB trapping, consistent with the reported ability of nucleosomal DNA to prime NF- κ B for binding (PMID: 34029641, PMID: 24086160, PMID: 15269206).

For Fig 3e, the consensus-based approach to determine the presence of the motif doesn't always work, a better approach would be to use a method as in Carter et al. LTR7 study.

Carter et al. uncovered subfamily-specific TFBS by first performing phyloregulatory analysis on the LTR7 biotype, revealing branch-specific regulatory features. While this represents a valuable

advancement, approaching all Alu and ERV subfamilies abundant in NFκB or IRF1 ChIP-Seq peaks in a similar manner will shift focus away from central questions, becoming a different story.

To overcome this, we have performed an additional scan of JASPAR 2022 PWM profiles in consensus sequences using FIMO tool applying two *Q-values*, $P < 1E-4$ for perfect match, and $P < 1E-3$ for permissive match. We find all interrogated NFκB motifs in top pTE contributors to NFκB binding. However, some were found with high confidence (*q-value* $P < 1E-4$), while others, such as RELs, with lower (*q-value* $P < 1E-3$), indicating the presence of proto-motifs only slightly deviating from their canonical PWM equivalents. IRF1 motif was also found in all ERV subfamilies, contributing the most to ChIP-Seq peaks.

Please note that we have excluded unrelated IRF motifs from the analysis, as corresponding ChIP-seq datasets were not explored. Consequently, assessing their presence in consensus sequences of pTEs bound by NF-κB or IRF1 is not informative.

Importantly, since all relevant motifs, or closely matching proto-motifs, were consistently identified in the consensus sequences of the interrogated subfamilies, we believe no additional analysis is required to support our conclusions.

For Fig 3f, was this analysis restricted to TFBS with peaks?

As the text states, it takes into consideration all TFBS within enhancers. In the revised version, we included the same analysis performed on motifs bound by the corresponding NFκB or IRF1 (revised Extended Data Fig. 5c and 6f). They demonstrate the same pattern.

For Fig 4, again it would be good to have a better control here. Could you perform the same analysis for other regions with Alus that were not found in the immune enhancers? Going back to the liftOver control I suggested above, it would be good to see if there's a difference but that control and these enhancer regions.

It is not yet understood how Alus may contribute to positive selection genome-wide. Given that they are among the most mutable elements in the human genome, including in enhancers (see our Fig.4c), their presence in randomly chosen regions may confer a selective advantage, a slight tendency we have observed genome-wide.

To rigorously test whether our Alu-derived enhancers are genuinely associated with elevated signals of positive selection, potentially via Alu-embedded SNPs, we expanded enhancers by ± 5 kb from their centers, thereby standardizing their size and incorporating adjacent genomic background.

Reassuringly, the same enrichment pattern persisted (Extended Data Fig. 7a), supporting the notion that enhancers with Alu have indeed a higher chance to be positively selected.

However, it was not true for enhancers potentially associated with inflammatory diseases (IDEs) from the original Fig.5d. Despite that we observed the tendency of IDE+background being more positively selected compared to non-IDEs+background, it did not reach statistical significance. Therefore, we removed the panel replacing it by the quantification of the fraction of positively selected Alu-derived SNPs in IDEs versus non-IDEs (see the revised Fig.5d, right panel). Similarly, we removed the corresponding sentence from the abstract.

Nevertheless, a higher proportion of IDEs carry SNPs with robust signal of positive selection ($P < 0.01$) compared to other enhancers and a higher fraction of SNPs are contained within Alus. This information is included in Fig.5.

Minor comments:

In Ext Fig 1a, could you add the number of regions in each column/row?

We have added the number of regions to the revised Ext Fig1d. Thank you for pointing this out.

Reviewer #2 (Remarks to the Author):

The authors investigated the evolutionary influence of transposable elements, and specifically Alu elements and endogenous retroviruses, on human immune-cell enhancers and inflammatory responses. Using comparative genomics, they traced enhancer sequence changes back to macaques and identified the role of transposons in reshaping NF- κ B and IRF1 binding motifs, particularly in great apes. They found that Alu-derived motifs often show increased binding affinity after the human-macaque split and are enriched for positive selection in humans, especially in enhancers associated with chronic inflammatory diseases. Their study highlights how transposon invasions uniquely shaped primate immune adaptation and continue to influence human inflammatory potential. The paper presents interesting and potentially compelling findings. However, to be accepted for publications, it requires several clarifications on how some of the analyses were performed, and on the rationale behind some of the analysis.

Specifically:

- Can the authors elaborate more about the immune-cell enhancers? Of the 60k enhancers, how any of them were present in all the cell types investigated? Or in how many cell types?

We thank the reviewer for the positive appreciation of our work and constructive comments and admit the initial lack of details. In the revised version, we are more explicit about the entire procedure in the main text and in the revised Extended Data Fig 1 and 2.

Initially, 60% of annotated regions were shared by at least two cell types. However, ATAC-seq profiles were more broadly shared among cell types (Extended Data Fig. 1b and d), indicating a broader degree of sharing than we initially anticipated.

Was there a threshold of number of immune cell types in which the enhancer was supposed to be detected? Or was one cell type enough?

A region was defined as “immune-cell enhancer” if it was active in at least one cell type. We clarified this in the text (see the first expanded paragraph the the first Results section). However, there are very few regions with individual cell type peaks (revised Extended Data Fig. 1d and below).

- Identification of pTE-derived enhancers: can the authors explain more in detail in the results section how much overlap between enhancers and TEs was required to affirm that an enhancer was TE derived? Is this based on the TE_ANALYSIS tool (so is this significant enrichment) or simple overlap? And in the latter case what fraction of the enhancer was required to overlap a TE?

We did not use the term “pTE-derived,” but rather “pTE-associated.” Since enhancer annotations are based on histone mark distributions, full coverage by pTE sequences is uncommon, Fig.1e (left panel) shows a median coverage of ~20%.

Enrichment is quantified by combining TE-ANALYSIS tool combined with the rigorous statistics (see further below). TE-ANALYSIS default parameters consider minimum 10bp overlap with TE. This is enough to capture significantly overrepresented subfamilies as we control for background. In the revised version, we explained this in more detail.

We employed the default pipeline of the TE_ANALYSIS tool, implementing a framework based on two complementary tests: a hypergeometric and a binomial tests for enrichment over background both by copy number and by length. Only adjusted p-values (Benjamini-Hochberg correction for multiple comparisons) were considered.

To control for potential bias from length distribution and liftOver, **we excluded**: (i) subfamilies enriched in 1,000 times shuffled random non-enhancer regions matched in number and length-distribution to each enhancer group; and now, in the revised version, (ii) subfamilies enriched in 1,000 times shuffled non-enhancer regions matched all enhancers and classified into three group based on the same liftOver strategy used for enhancer classification (new control required by the Reviewer #1).

The remaining subfamilies significantly enriched in each enhancer group are listed in Supplementary Dataset 3.

A few sentences below they mention that the pTEs contribute to 2-3% of the length of the enhancer. But given that TEs represent half of the human genome, wouldn't this be significantly less than expected by chance?

We believe there may be a misunderstanding. In Fig. 1e (left panel), we report that pTEs cover between 10% and 100% of enhancer length, with a median exceeding 20% for intermediate enhancers.

In contrast, the 2-3% coverage cited by the reviewer refers specifically to pTEs gained after the human-macaque split, not to all pTEs. Given their recent origin, this modest coverage is expected. Importantly, despite their limited coverage, these recently acquired insertions disproportionately contribute to the

novel accessible chromatin and functional TFBS, including a significant fraction of great-ape-specific binding sites. In this context, their regulatory impact may outweigh their physical footprint.

While it is true that transposable elements comprise roughly half of the human genome, this includes both ancient and primate-specific TEs. Our analysis focuses on primate-specific elements, which together cover only ~22.8% of the genome.

Moreover, our focus has not been on the overall enrichment of all pTEs in enhancers relative to the genome, but rather the selective enrichment of specific subfamilies, each with distinct phylogenetic origin and regulatory motifs, suggesting possible adaptive recruitment.

In the methods they mention a 90% overlap required for the gap analysis (and 50% for the ATAC-seq peaks), but what about all other comparisons? I am not implying that the analysis was not performed correctly, but simply that the result section would benefit from much more detailed description of the approach.

We thank the reviewer for raising this point. This information has now been clarified and is explicitly stated at every relevant point in the revised manuscript.

- Again, in the sentence: Over 70% of dynamic regions overlapped with pTEs from these enriched subfamilies, compared to less than 50% of static regions, what fraction of the length of the enhancer was set as threshold for being defined as “overlapping a TE”?

As mentioned above, we followed the default setting of the TE-ANALYSIS pipeline, which defines overlaps at a minimum of 10 bp, sufficient to capture meaningful enrichment while minimizing background noise. We also apologize for the earlier inaccuracy: the correct value is 69%, not 70%, and this has been corrected in the revised manuscript.

- On page 7, the authors write that Alu elements are the most abundant transposons in the human genome, but that is incorrect. LINE elements are, by far, the most abundant.

The reviewer is correct that LINEs occupy a greater proportion of the genome by total coverage. Our intention, however, was to emphasize that Alu elements are the most abundant *primate-specific* TEs in terms of copy number (1,180,685 Alu copies versus 213,368 primate-specific LINEs). We acknowledge the omission of “primate-specific” in the original text and have now corrected this for clarity.

- Why did the authors focus on Alu and ERVs, not considering LINEs and SVAs? SVAs in particular are considered to be a major source of human-specific enhancers. Could they provide more rationale on the choice of investigate TEs?

Indeed, both SVAs and LINES can contribute to human enhancers and harbor transcription factor binding motifs. However, our study specifically focused on Alus and ERVs, which represent the most abundant sources of great-ape-specific, inflammation-related TFBS.

We do not intend to discount the relevance of SVAs or LINES for IRF1 or NF- κ B binding. Rather, Alus and ERVs have been more consistently linked to these factors in the literature and emerged as dominant contributors in the analysed datasets. We have now clarified this rationale in the revised Results section.

**- Page 9: the sentence “with gibbon added as a lesser ape to identify great-ape-specific gains”.
Could the authors clarify what this means?**

We agree that the original phrasing may have lacked clarity. Gibbon, classified as lesser ape, occupies an intermediate evolutionary position between macaques and the great apes. We included the gibbon genome to more precisely infer the timing of additional TFBS emergence within enhancers following macaque divergence (before or after great apes). This clarification has been incorporated into the revised manuscript.

- When the authors state that “pTEs made significant contributions to “shared” TFBS, accounting for 10% to 50% of the sites”: how does this compare to random expectation? Is it more than expected by chance? Was there a p-value calculated?

We are not entirely sure what the reviewer means by “random expectation” in this context. All TFBS shown in Fig. 3a are significantly enriched within enhancers and ATAC-seq peaks (see revised Supplementary Dataset 4). Figure 3a simply categorizes these enriched TFBS by their sequence origin.

We expect NF- κ B-related motifs to be widespread across genomic Alus, as they are detectable in the consensus sequences of the most abundant Alu subfamilies. What distinguishes enhancer-linked Alu-derived NF- κ B motifs (and pERV-derived IRF1 motifs), however, is their evolutionary refinement: these motifs exhibit higher predicted binding affinity compared to their counterparts in consensus sequences and, by extension, to Alu-derived motifs across the broader genome (Fig. 3g).

- Regarding the CHIP-seq publicly available datasets (ENCODE) that the authors used: what was the read length? Were the reads paired ends? Were only uniquely mapped reads retained? These are key parameters to be implemented to be able to properly map reads on TE regions .

The reviewer is correct to raise these concern. Among CHIP-seq datasets we used, only around 25% were sequenced in paired-end. Read length varies from 50 to 100nt. However, the standardized ENCODE pipeline requires only uniquely mapped reads, which addresses the mappability issue but may underestimate reads on Alus.

- Based on the methods, it seems that the authors did not correct for multiple testing (i.e no FDR) in their enrichment analysis performed with the TE analysis pipeline tool. Could the authors elaborate on why?

As stated above, we employed the default pipeline of the TE_ANALYSIS tool, implementing two complementary tests: a hypergeometric and a binomial tests for enrichment over background both by copy number and by length. Only adjusted p-values (Benjamini-Hochberg correction for multiple comparisons) were considered. These values were given in the original Supplementary Dataset 3. Therefore, we corrected for multiple testing and used very robust statistics. However, we definitely failed to be explicit in our Methods sections, which is now fixed.

- In the methods pLINA should be pLINE?

It is corrected. We thank the reviewer for noticing this.

Reviewer #3 (Remarks to the Author):

This study from Elina Zueva and colleagues finds an important and innovative evolutionary details of primate endogenous retroviruses (pERVs) and Alu elements. pERVs are remnants of ancient viruses that infiltrated the genomes of our primate ancestors and are now 8% of human genome. Initially, these viral invaders behaved like parasites, but over time they seeded our DNA with special sites that could attract immune regulators such as the IRF1 protein. In contrast, Alu elements, tiny, highly abundant DNA sequences (1 million copies) that make upto 10 % of our genome have evolved to favor the binding of NF- κ B, a protein that is crucial in managing inflammation. Although a single Alu element might have a small effect, the cumulative influence of thousands of them can dramatically impact how our genes respond to stress or infection. One of the novel findings of this research is the discovery that these elements are not static relics; they have been continuously refined by evolution. In particular, the study shows that specific segments within Alu elements have evolved precise NF- κ B binding motifs. For example, a distinct Alu-derived NF- κ B1 motif (referred to as MA0105.4) underwent rapid expansion in the common ancestor shared by humans and chimpanzees. This rapid expansion suggests that these elements provided ready-made genetic tools during periods of rapid environmental change, allowing our ancestors to fine-tune their inflammatory responses swiftly and effectively. This adaptive mechanism, forged over millions of years, not only explains the rapid evolution of our inflammatory responses but may also account for the genetic basis of susceptibility to inflammatory diseases in modern humans. Ultimately, these ancient DNA elements continue to contribute to our resilience by providing a vast, dynamic reservoir of genetic variability that can be drawn upon to meet new challenges.

I like this study as it gets close to answer why there is a remarkable proliferation of Alu elements in

the primate genome. It suggests that these elements may have undergone positive selection driven by their potent enhancer activity during immune cell stimulation, thereby contributing significantly to the host's regulatory responses upon pathogen invasion.

This work indeed represents a significant advance over previously published studies. Thus, this line of advancement has enormous potential to outreach plenty of genome biology researchers, chromatin and transposon audiences.

Overall, although preliminary, the results are interesting and worthy of publishing. At this point, however, several issues arise that need further clarification and analysis before I consider this study complete and make me less optimistic about this work getting published in its current form.

We thank the reviewer for the enthusiasm regarding the manuscript and their constructive comments and suggestions.

Major Concerns:

1. My major concern is that the manuscript does not adequately address its limitations. For example, the authors use activity-by-contact (ABC) maps to infer looping interactions between TE-derived enhancers and gene promoters. However, these maps only show physical proximity and do not establish a functional regulatory relationship. Moreover, it is unclear which specific genes are implicated and how these genes contribute to the innate immune response. A more thorough discussion identifying the gene sets and either referencing existing functional data or proposing targeted experiments would greatly strengthen the manuscript's premise. With functional data, I mean genetic experiments where the study has rescued the phenotype e.g., MER41B from PMID: 26941318.

We used the Activity-by-Contact (ABC) map because it has recently been demonstrated to be among the most reliable algorithms for identifying enhancer-gene pairs (PMID: 31784727). Rather than being based on the proximity, it employs very complex algorithm. We quote: "*This (ABC) model is based on the simple biochemical notion that an element's quantitative effect on a gene should depend on its strength as an enhancer ("Activity") weighted by how often it comes into 3D contact with the promoter of the gene ("Contact"), and that the relative contribution of an element on a gene's expression (as assayed by the proportional decrease in expression upon CRISPR-inhibition) should depend on the element's effect divided by the total effect of all elements.*" Fulco et al. performed extensive perturbation assays to demonstrate that their model substantially outperforms previous methods at predicting the complex connections.

To identify genes implicated in immune response, we used ABC predictions and genes upregulated during immune stimulation of human whole blood cells from *Hawash et al.* (PMID: 33771921). These results have been illustrated in the original version of the manuscript. However, we largely underexplored these genes. Following the reviewer's advice, we performed a deeper analysis (revised Extended Data Fig.5 and later in more detail).

Regarding referencing of existing functional evidence, we initially cited *Liang et al.* (PMID: 37438529), which showed that CRISPR-Cas9-mediated deletion of Alus within enhancers altered target gene expression. In the Discussion section, we noted that many Alus identified in our enhancer set overlap with those validated in their perturbation screen. These Alus, along with their corresponding enhancer coordinates, were listed in the original Supplementary Dataset 9. We have revised the phrasing to more clearly highlight this overlap, which strengthens our conclusions, even though the functional evidence derives from the perturbation assays reported by the other group.

2. Page 7, last sentence: The authors suggest that Alu insertions drive the formation of open chromatin. However, the evidence supporting this claim remains ambiguous. It is equally plausible that Alu elements preferentially integrate into preexisting open chromatin regions rather than creating novel enhancers. In fact, Extended Data Figure 2C appears to support this alternative interpretation, showing that ATAC-seq signals are similarly distributed in the neighboring regions of Alu insertions. A more detailed discussion of these observations to distinguish between these scenarios, would significantly strengthen the authors' conclusions.

The original Extended Data Fig. 2c illustrated genome tracks with a peak summit within the novel Alu integrant. The coincidence of the maximal signal with this Alu's body strongly suggests that this element created novel accessibility hotspot.

To better support this interpretation, we visualized ATAC-seq signal density over Alu and pERV elements that contribute $\geq 50\%$ of the ATAC-seq peak length and were part of the quantification shown in the original Extended Data Fig. 2 (now in revised Extended Data Fig. 3c).

To address, however, the reviewer's concern, we now present these findings more cautiously in the manuscript, framing this as a hypothesis rather than a definitive conclusion.

3. On Page 10, Authors declare the Modus operandi that these enhancers regulating conserved and inflammation aspects of immune response - a claim that currently rests solely on computational analyses. Such a substantial conclusion warrants further functional validation. Specifically, experiments such as CHIP-STARR-seq would be necessary to confirm the enhancer activity of these Alu elements, followed by genetic perturbation studies to demonstrate that disruption of these enhancers leads to dysregulation of target gene expression and impacts the

associated phenotype. I do not recommend that the authors should provide additional experimental evidence but they should tone down the claim.

We thank the reviewer for pointing this out. We have now tempered our conclusions and complemented them with a more in-depth analysis of gene sets predicted as targets of our enhancers by ABC maps (see further below).

4. Page 11 states that TE enhancers are entirely embedded within the corresponding ChIP-seq peaks which is the basis for selecting these enhancers. However, it remains unclear whether these enhancers overlap specifically with the peak summits and whether the sequences flanking these summits are enriched for the expected transcription factor binding motifs. I recommend that the authors provide a detailed analysis of the spatial distribution of TE enhancers relative to the ChIP-seq peak summits and assess motif enrichment in these critical regions. This additional information would substantially bolster the evidence for their proposed functional role.

Given that Fig.3 is dedicated to specific motifs, in page 11, we stated “*motifs are fully embedded within ChIP-seq peaks*”, while peak summits should coincide with our enhancer regions (now mentioned in the main text).

To answer the question of spatial distribution, we visualized the distribution of NF- κ B ChIP-Seq signal with respect of Alu and ERV-derived motifs in four different lymphoblastoid cell lines (revised Fig. 3d and below).

In most cases, both Alu- and pERV-derived NF- κ B motifs coincide with high ChIP-seq signal (compared to flanks), indicating robust *in vivo* binding. Overall, Alu-derived motifs exhibit somewhat broader signal distribution than those from pERVs, potentially reflecting a distinct regulatory behavior. It is possible that a subset of Alu-derived TFBS functions in NF- κ B trapping or buffering, in line with prior evidence that nucleosomal DNA can facilitate NF- κ B priming and retention (PMID: PMID: 34029641; PMID: 24086160; PMID: 15269206).

5. Page 18: Authors write and claim “enhancers enriched in Alus suggest their key role in the

evolutionary shaping of the human inflammatory response”. This is a provocative claim based on evidence of correlation from neighbor genes and gene ontology is substandard in the field. The investigations presented here are exploratory. Authors should refrain from making bold conclusions based on association analysis of TEs and genes and/or biological pathways unless the association is already published with a robust, reliable and reproducible set of experiments. I would encourage authors to re-write these sections with keeping the limitations of the gene ontology (DisGeNet in this case) and correlational tools in their conclusions.

We thank the reviewer for highlighting the boldness of this claim, which we intended, but failed to present as a mere hypothesis. We have now revised the wording to soften the conclusion, explicitly acknowledging the limitations of our study.

6. The reanalysis of data on Page 31: The manuscript relies on ATAC-seq data from Corces et al. (Nat. Gen. 2016), which derives from selected and some cancer cell types, yet it remains unclear why the dataset from Calderon et al. (Nat. Gen. 2018) which encompasses ATAC-seq and RNA-seq profiles in the atlas form for diverse human T and B cell types in both resting and activated states was not incorporated. Given the physiological relevance of the Calderon dataset for assessing chromatin accessibility in immune cells, its inclusion could potentially provide a more robust validation of the observed enhancer dynamics. I recommend that the authors clarify their rationale for excluding this dataset or consider integrating these data to further substantiate their conclusions.

We are grateful the reviewer noticed this discrepancy. We did, of course, analyze the ATAC-seq data from *Calderon et al.*, as indicated in the original Supplementary Dataset 1. However, due to an oversight, only the study by *Corces et al.* was cited in both the main text and the Methods section. We regret this omission and have now corrected the citations accordingly. Notably, both *Calderon et al.* and *Corces et al.* datasets are used in analysis provided in the revised Extended Data 1 and 3.

7. Page 32: Regarding target gene selection, it is crucial to distinguish whether a given enhancer functions as the sole regulatory element for its target gene or operates alongside multiple enhancers. Genes often possess multiple enhancers, including primary and shadow enhancers, which can act redundantly to ensure robust gene expression. Therefore, attributing gene regulation to a single enhancer without considering the broader enhancer landscape may lead to oversimplified conclusions. I recommend that the authors assess the enhancer redundancy within the genomic context of each target gene to accurately interpret the regulatory mechanisms involved.

We fully agree that enhancers can act redundantly. We appreciate this thoughtful suggestion and have now performed the proposed analysis to delve deeper into the regulation of the immune response gene by static, intermediate and rapidly evolving enhancers. Using ABC-predictions for gene upregulated during immune response, we observed a higher number of intermediate enhancers per gene (median of five) compared to other groups (Extended Data Fig.4b and below). We then identified two similarly sized gene groups with an intermediate-to-static enhancer ratio either above or below one (Extended Data Fig.4c). Genes with a higher ratio (biased toward intermediate enhancers) were more enriched in immune-related terms, including inflammation, by both *P-value* and gene count per term (Extended Data Fig. 4d and below), suggesting that intermediate enhancers may increase the flexibility and redundancy of immune response regulation and supporting our initial hypothesis. We appreciate this meaningful proposal to perform this analysis.

8. Methodology on Page 33: The authors analyze the enrichment of TE subfamilies within enhancer regions by normalizing the length of these TEs but do not address how TE length may influence this enrichment. Longer TEs, such as full-length proviruses (~7 kb), might be underrepresented in enrichment analyses compared to shorter elements like solo long terminal repeats (LTRs) or Alu elements, which are only a few hundred base pairs long. This size discrepancy could lead to a bias favoring the detection of shorter TEs as enhancers. While I do acknowledge that it is essential to consider the impact of TE length on enrichment analyses, the longer TEs e.g., LTR12, HERVH, etc., that have a strong cis-regulatory potential especially have antiviral activities might be underrepresented in the analysis.

The reviewer is correct that TE enrichment could be influenced by TE length. We did, of course, account for this in our initial analysis, but unfortunately did not detail it in the text corresponding to Fig.1. However, the relevant statistical results were included in the submitted initially Supplementary Dataset 3, where the columns titled “*binom. adj. P-value by length*” and “*hypergeometric adj. P-value by length*” reflect the enrichment of each TE subfamily while controlling for its total length in the genome. We have clarified this in the main text.

Minor Comments:

1. Authors should discuss why they did not observe THE1B elements with NFκB binding and cite the relevant literatures. THE1B was shown to have high chromatin accessibility in T-cells with NFκB motif and binding (PMID: 30381291, PMID: 39988678). Of note, PMID: 39988678 is my paper and I do not encourage authors to cite if they do not reason its relevance.

We observe NF-κB binding to THE1B elements in ChIP-seq datasets. It was the most abundant of ERV subfamilies in NF-κB CHIP-Seq peaks, providing numbers of motifs comparable to that of Alus. Unfortunately, this observation was not highlighted in the first version of the manuscript, as our initial analysis focused on Alus providing the most of NF-κB motifs within ChIP-seq peaks and ERVs providing the most IRF1 motifs to IRF1 ChIP-Seq peaks. We have now reorganised Fig.3 to make it more logical, moving IRF1 to the revised Extended Data Fig.6 and focusing on NFκB in the main figure. We also included THE1B as, rightfully, this is the most important contributor to NFκB binding compared to other pERVs. We also quote PMID: 39988678 as it states that THE1B participates in antiviral response and, therefore, may be linked to inflammation.

Figure 1B: Are the differences statistically significant ?

It is very significant. See the revised Fig.1 and the one below.

b Conservation across
17 primate genomes

Figure 1 C: Why do authors not perform a comparative analysis of enhancer regions with heterochromatin regions in pairwise manner which might help them calculating a relative frequency.

Fig. 1c aims to compare mutation rates between static and dynamic enhancers within sequences that are alignable to the macaque genome without gaps. In this context, the enhancer groups serve as internal controls for one another, allowing a direct comparison of evolutionary dynamics. We believe that including heterochromatic regions would not provide an appropriate reference here, as their mutation rates are shaped by fundamentally different chromatin contexts and selective constraints.

Figure 1F: Labels on Y-axis (the name of TE families) looks cluttered. You can decrease the font to make them more visible.

De-cluttered now. Thank you.

Page 7, line 14: Please do not pitch the lines as less abundant TEs are more regulatory from a review journal. It might be more likely that host would conserve few copies with strong regulatory activities but there are remarkable exceptions including the one being shown in this paper.

We agree. The text has now been revised.

Page 7, line 22: The word co-option is not appropriate here. To show co-option, you would require a genetic and rescue experiments as I have mentioned above.

We acknowledge that the original phrasing was not appropriate and revised it.

We thank you and the reviewers for the positive evaluations. We have carefully addressed all remaining points as outlined below.

REVIEWERS' COMMENTS

Reviewer #1 (Remarks to the Author):

The detailed response from the authors has helped address all my concerns. A lot of new controls have now been added and confirm the robustness of the results.

In particular, I do appreciate the additional controls that were included to show the impact of the size of the enhancers and of the leftover step. A small point but my interpretation of the new Fig S1f is that changing the size of the enhancers does have a pretty significant effect, since it changes the classification of 25%-33% of the "inter" and "rapid" category. My suggestion would be to present this result in a more balanced way. I don't quite agree with "... yielded comparable classification patterns, indicating that the merging procedure did not introduced any significant bias".

We thank the reviewer for the positive feedback. The interpretation of Fig. S1f is now more balanced: "A similar analysis using length-normalised ($\pm 1,000$ bp from midpoints) or individual cell type enhancers yielded **overall** comparable classification patterns, indicating that the merging procedure **did not substantially biased the analysis**".

We prefer to use "overall comparable," as it acknowledges some differences while indicating that the patterns remain globally comparable.

Reviewer #2 (Remarks to the Author):

The authors have addressed all my concerns, the manuscript can be accepted for publication.

We thank Reviewer #2 for the positive evaluation and for recommending acceptance of our manuscript.

Reviewer #3 (Remarks to the Author):

The authors have adequately addressed my comments, and the manuscript is much improved.

We thank Reviewer #3 for the positive evaluation and for recommending acceptance of our manuscript